# A ribosome assembly stress response regulates transcription to maintain proteome homeostasis

Benjamin Albert[1], Isabelle C Kos-Braun[2], Anthony K Henras[3], Christophe Dez[3], Maria Paula Rueda[1], Xu Zhang[1], Olivier Gadal[3], Martin Kos[2], David Shore[1]*

[1]Department of Molecular Biology, Institute of Genetics and Genomics of Geneva (iGE3), Geneva, Switzerland; [2]Heidelberg University Biochemistry Center (BZH), Heidelberg, Germany; [3]Centre de Biologie Intégrative, Université Paul Sabatier, Toulouse, France

**Abstract** Ribosome biogenesis is a complex and energy-demanding process requiring tight coordination of ribosomal RNA (rRNA) and ribosomal protein (RP) production. Given the extremely high level of RP synthesis in rapidly growing cells, alteration of any step in the ribosome assembly process may impact growth by leading to proteotoxic stress. Although the transcription factor Hsf1 has emerged as a central regulator of proteostasis, how its activity is coordinated with ribosome biogenesis is unknown. Here, we show that arrest of ribosome biogenesis in the budding yeast *Saccharomyces cerevisiae* triggers rapid activation of a highly specific stress pathway that coordinately upregulates Hsf1 target genes and downregulates RP genes. Activation of Hsf1 target genes requires neo-synthesis of RPs, which accumulate in an insoluble fraction and presumably titrate a negative regulator of Hsf1, the Hsp70 chaperone. RP aggregation is also coincident with that of the RP gene activator Ifh1, a transcription factor that is rapidly released from RP gene promoters. Our data support a model in which the levels of newly synthetized RPs, imported into the nucleus but not yet assembled into ribosomes, work to continuously balance Hsf1 and Ifh1 activity, thus guarding against proteotoxic stress during ribosome assembly.

DOI: https://doi.org/10.7554/eLife.45002.001

*For correspondence:
David.Shore@unige.ch

**Competing interests:** The authors declare that no competing interests exist.

## Introduction

Ribosome assembly is the most energy demanding process linked to cell growth and requires coordinated production of processed ribosomal RNAs (rRNAs), ribosomal proteins (RPs) and ribosome biogenesis (RiBi) factors. This massive biosynthetic program permits rapidly growing yeast cells to produce about 2000 ribosomes per minute (*Warner, 1999*), which is critical for sustaining high rates of growth (mass accumulation) and proliferation. At the same time, though, ribosome assembly poses a constant threat to cellular protein homeostasis and continued growth, since it requires the coordinated and large-scale assembly of four rRNAs with 79 different RPs, the latter of which are known to be highly prone to aggregation (*David et al., 2010*; *Pillet et al., 2017*; *Rand and Grant, 2006*; *Weids et al., 2016*). Indeed, unassembled RPs in metazoans have long been known to activate p53, through titration of its negative regulator MDM2, and conserved p53-independent pathways that respond to perturbations in ribosome assembly are now beginning to emerge (*James et al., 2014*). Given the absence of p53 in yeasts, *Saccharomyces cerevisiae* is a promising model system in which to uncover ancestral processes that might monitor ribosome assembly to regulate growth and protein homeostasis in eukaryotes.

Heat shock factor 1 (Hsf1) is a central actor in Protein Quality Control (PQC) and protein homeostasis (proteostasis) in eukaryotes, in both stressed and unstressed cell, and in pathological situations

**eLife digest** When yeast cells are growing at top speed, they can make 2,000 new ribosomes every minute. These enormous molecular assemblies are the protein-making machines of the cell. Building new ribosomes is one of the most energy-demanding parts of cell growth and, if the process goes wrong, the results can be catastrophic. The proteins that make up the ribosomes themselves are sticky. Left unattended, they start to form toxic clumps inside the compartment that houses most of the cell's DNA, the nucleus.

A protein called Heat shock factor 1, or Hsf1 for short, plays an important role in the cell's quality control systems. It helps to manage sticky proteins by switching on genes that break down protein clumps and prevent new clumps from forming. Hsf1 levels start to rise whenever cells are struggling to keep up with protein production. If it is half-finished ribosomes that are causing the problem, cells can stop making ribosome proteins. The protein in charge of this in yeast is Ifh1. It is a transcription factor that sits at the front of the genes for ribosome proteins, switching them on. When yeast cells get stressed, Ifh1 drops away from the genes within minutes, switching them off again. Yet how this happens, and how it links to Hsf1, is a mystery.

To start to provide some answers, Albert et al. disrupted the production of ribosomes in yeast cells and examined the consequences. This revealed a new rescue response, that they named the "ribosome assembly stress response". Both Hsf1 and Ifh1 are sensitive to the build-up of unfinished ribosomes in the nucleus. As expected, Hsf1 activated when ribosome proteins started to build up, and switched on the genes needed to manage the protein clumps. The effect on Isfh1, however, was unexpected. When the unassembled ribosome proteins started to build up, it was the clumps themselves that pulled the Ifh1 proteins off the genes. The unassembled ribosomes proteins seemed to be stopping their own production. Low levels of clumped ribosome proteins in the nuclei of unstressed cells also helped to keep Hsf1 active and pull Ifh1 off the ribosome genes. It is possible that this provides continual protection against a toxic protein build-up.

These findings are not only important for understanding yeast cells; cancer cells also need to produce ribosomes at a very high rate to sustain their rapid growth. They too might be prone to stresses that interrupt their ribosome assembly. As such, understanding more about this process could one day lead to new therapies to target cancer cells.

DOI: https://doi.org/10.7554/eLife.45002.002

---

(*Li et al., 2017*). Notably, Hsf1 is a direct modulator of tumorigenesis and becomes essential, as it is in budding yeast (*Solís et al., 2016*), to support growth of malignant cells (*Santagata et al., 2013*). Hsf1 prevents protein aggregation and proteome imbalance by driving the expression of a small regulon including genes encoding essential chaperones (Hsp70/Hsp90), nuclear/cytoplasmic aggregases, and proteasome components (*Solís et al., 2016*; *Mahat et al., 2016*; *Pincus et al., 2018*). Interestingly, studies in budding yeast reveal that the Ribosome Quality Control complex (RQC), conserved from yeast to human (*Brandman et al., 2012*), increases Hsf1 activity under conditions of translation stress. However, many essential aspects of Hsf1 regulation remain to be elucidated, in particular whether its transcriptional activity is linked to ribosome biogenesis itself. Recently, a conserved PQC mechanism referred to as Excess Ribosomal Protein Quality Control (ERISQ) was described that specifically recognizes unassembled RPs in the nucleus and targets them for proteasome degradation (*Sung et al., 2016a*; *Sung et al., 2016b*), thus illuminating observations made 40 years ago showing that excess RPs are rapidly degraded (*Gorenstein and Warner, 1977*; *Warner, 1977*). Sung and colleagues showed that the ubiquitin ligase Tom1 plays an important role in ESRIQ by preventing the accumulation of detergent-insoluble RPs. The potential role of Hsf1 in ERISQ has not yet been explored.

Given the tremendous investment of cellular resources involved in ribosome production (*Warner, 1999*) and the fact that a decrease of ribosome abundance protects cells against proteotoxic stress (*Guerra-Moreno et al., 2015*; *Mills and Green, 2017*), it might be expected that cells have evolved mechanisms to rapidly decrease RP gene transcription in the face of defects in ribosome assembly, in order to both save energy and reestablish cellular proteostasis. In *S. cerevisiae*, RP gene transcription is known to be tightly regulated according to growth conditions through the

stress-sensitive transcription factor (TF) Ifh1. Thus, Ifh1 is rapidly released from RP promoters only minutes following inhibition of the conserved eukaryotic growth regulator Target Of Rapamycin Complex 1 (TORC1) kinase (*Schawalder et al., 2004*). Although it has been shown that Ifh1 promoter binding is coordinated with RNA polymerase I (RNAPI) activity upon prolonged TORC1 inhibition to help balance RP and rRNA production (*Albert et al., 2016*; *Rudra et al., 2007*), how Ifh1 is removed from RP gene promoters to immediately downregulate their expression following stress remains a mystery. Furthermore, possible links between RP gene expression, ribosomal assembly and the protein homeostasis transcription program driven by Hsf1 remain important open questions.

In this study, we uncover a novel regulatory pathway, hereafter referred to as the Ribosomal Assembly Stress Response (RASTR), that allows rapid and simultaneous up-regulation of protein homeostasis genes and downregulation of RP genes following disruption of various steps in ribosome biogenesis (rRNA production, processing or RP assembly). We show that RASTR is highly specific to the RP and Hsf1 regulons, with little or no effect on a much larger group of genes implicated in the Environmental Stress Response (ESR). Importantly, RASTR requires neo-synthesis of RPs following stress and is linked to the accumulation of RP aggregates, which we propose lead to Hsf1 activation, through chaperone competition, and to the sequestration of Ifh1 in an insoluble nucleolar fraction. Notably, we show that protein synthesis inhabitation via cycloheximide treatment leads to a transcriptional response opposite to that of RASTR, supporting a model in which unstressed cells constantly monitor nuclear levels of unassembled RPs and use this information to balance expression of Hsf1 target genes with those encoding RPs. Finally, we demonstrate that RASTR is the initial transcriptional response to inactivation of TORC1 kinase, supporting a key role for this regulatory pathway in the activation of a protein homeostasis transcriptional program that allows cells to cope with the proteotoxic consequences of disruptions to ribosome biogenesis.

## Results

### Topoisomerase depletion triggers a rapid repression of RP genes and activation of proteostasis genes

In an effort to better understand the role of the two major eukaryotic DNA topoisomerases in protein-coding gene transcription, we generated yeast strains in which Top1, Top2, or both of these enzymes are rapidly degraded by the auxin-induced degron (AID) method (*Nishimura et al., 2009*) and confirmed by western blotting that significant depletion of either protein was obtained between 10 and 20 min following auxin addition to the medium, and that Top2 depletion, as expected, prevents cell growth (*Figure 1—figure supplement 1A,B*). We then performed ChIP-seq analysis of RNA polymerase II (RNAPII) in the Top1-AID, Top2-AID and Top1/2-AID strains at 20 and 60 min following auxin addition (*Figure 1A–C*, *Supplementary file 1*). As expected (*Brill et al., 1987*; *Brill and Sternglanz, 1988*), the absence of Top2 had little or no effect on RNAPII distribution (*Figure 1B*). However, Top1 depletion triggered a rapid response at two specific groups of genes: upregulation of Hsf1 target genes and downregulation of RP genes (*Figure 1A,D*; *Figure 1—figure supplement 1C,D*). Remarkably, this response was transient, as both groups of genes returned to normal levels (i.e. before auxin addition) by 60 min. This re-equilibration was dependent upon Top2 since it failed to occur in the Top1/2-AID strain, where prolonged auxin treatment led to significant dysregulation of many other RNAPII-transcribed genes (*Figure 1C*).

Since upregulation of proteostasis-related genes and downregulation of RP genes are characteristic of many different stress responses, we decided to quantify the effect of Top1 depletion on transcription all gene groups that have been classified as part of the general 'Environmental Stress Response' (ESR; *Gasch et al., 2000*), which include an additional group of stress-induced genes regulated by the Msn2/4 TFs and a large suite of genes involved in ribosome biogenesis (RiBi genes). This analysis shows clearly that Top1 depletion, as well as depletion of both Top1 and Top2 (at the early 20 min time point), triggers a highly specific stress response linked to RP genes and Hsf1 target genes (*Figure 1A–C*). Such a targeted response is unlikely to result from a global topological effect on RNAPII recruitment but would instead appear to be the consequence of the activation of a specific signaling pathway that is more restricted in nature than the ESR.

To explore the target(s) of this hypothetical signaling pathway at RP genes, we monitored by qPCR ChIP the promoter association of three TFs (Rap1, Fhl1 and Ifh1) that operate at the majority

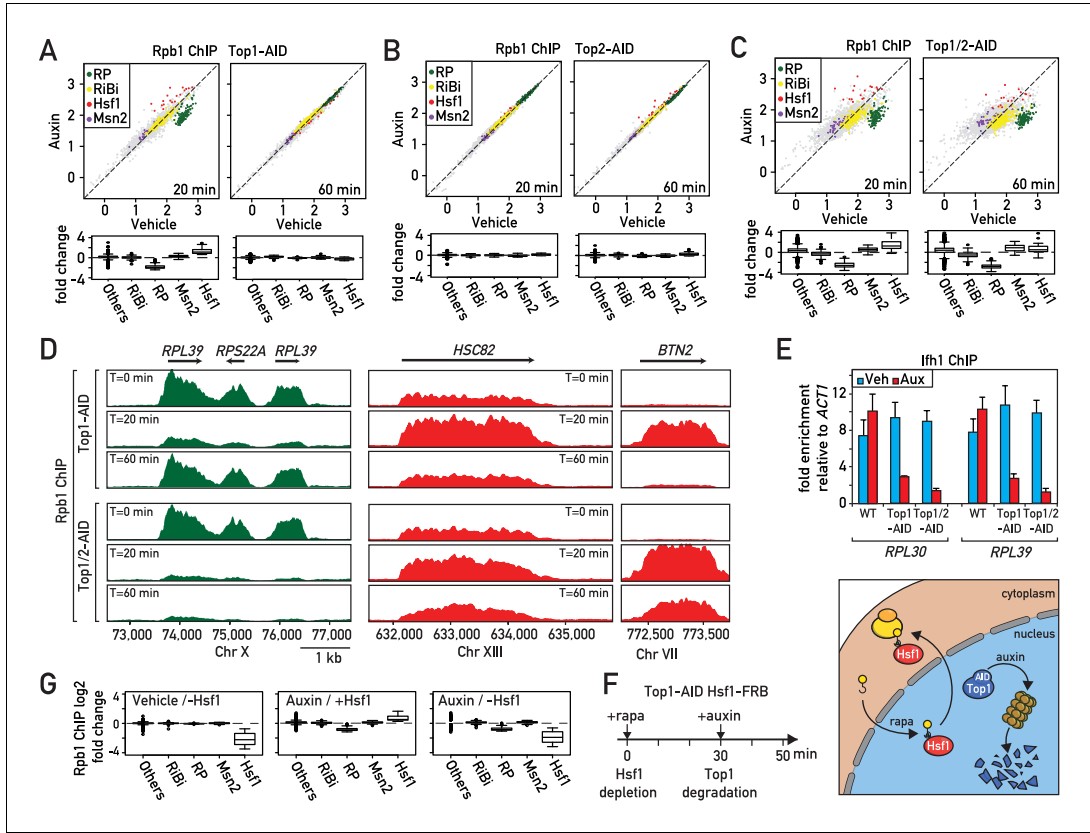

**Figure 1.** Rapid degradation of Topoisomerase 1 (Top1) induces a transient induction of Heat Shock Factor 1 (Hsf1) target genes and down-regulation of ribosomal protein (RP) genes. (**A, B, C**) Scatter plots (top panels) and box plots (bottom panels) comparing RNAPII binding (as measured by Rpb1 ChIP-seq) in Top1-AID (**A**), Top2-AID (**B**) and Top1/2-AID (**C**) strains at the indicated times following either auxin or vehicle addition to the media. Indicated gene categories (RP, n = 138; ribosome biogenesis [RiBi], n = 229; Msn2 target genes, n = 45; and Hsf1 target genes, n = 19) are color-coded on the scatter plots and displayed separately on the box plots, together with all remaining genes as a fifth class (others, n = 4610). (**D**) Genome browser tracks showing RNAPII (Rpb1) ChIP-seq read counts at the indicated positions on chromosomes X, XIII and VII at 0, 20, or 60 min following auxin addition to Top1-AID (top panels) and Top1/2-AID (bottom panel) strains. Gene names and open reading frame (ORF) positions are shown above. (**E**) Ifh1 occupancy, measured by qPCR ChIP at the *RPL30* and *RPL39* promoters 20 min following auxin addition to cultures of Top1-AID and Top1/2-AID strains. Bar height indicates the average and error bars the range of n = 4 biological replicates. (**F**) Schematic representation of protocol for Hsf1-FRB nuclear depletion (anchor-away) induced by rapamycin (Rapa) followed by Top1-AID depletion. (**G**) Box plots showing RNAPII (Rpb1) ChIP-seq signal following Hsf1-FRB nuclear depletion by anchor-away (-Hsf1, left panel), Top1-AID degradation (auxin, middle panel) or both Hsf1-FRB and Top1-AID depletion (auxin / -Hsf1, right panel) for the five functional groups described in (**A**).

DOI: https://doi.org/10.7554/eLife.45002.003

The following figure supplement is available for figure 1:

**Figure supplement 1.** Characterization of Top1-AID, Top2-AID, Top1-AID Top2-AID and Top1-AID Hsf1-FRB strains.

DOI: https://doi.org/10.7554/eLife.45002.004

(>90%) of these 138 genes (**Knight et al., 2014**). Interestingly, we found that the activator Ifh1 is rapidly released from RP gene promoters after topoisomerases depletion (**Figure 1E**), whereas Rap1 and Fhl1, which bind directly to RP promoter DNA, are not affected (**Figure 1—figure supplement 1E–F**). To confirm that Hsf1 is indeed required for upregulation of genes following Top1 depletion, we used the anchor-away technique (**Haruki et al., 2008**) to rapidly remove Hsf1 from the nucleus (30 min treatment with rapamycin; **Solís et al., 2016**) before initiating Top1-AID degradation by auxin addition (**Figure 1F**). Efficient nuclear depletion of Hsf1 was confirmed by the inability of the

Hsf1-FRB, Top1-AID strain to form colonies in the presence of rapamycin (*Figure 1—figure supplement 1G*). Note that the strain used in this and all other anchor-away experiments contains the *TOR1-1* mutation and is thus resistant to the normal physiological consequences of rapamycin treatment, which inactivates the growth-promoting TORC1 kinase (*Heitman et al., 1991*; *Loewith and Hall, 2011*). This experiment revealed that Hsf1 nuclear depletion completely abolishes activation of stress genes following Top1 depletion without affecting down-regulation of RP genes (*Figure 1G*). Therefore, activation of stress-induced genes following Top1 depletion is completely Hsf1-dependent, whereas repression of RP genes is independent of Hsf1 or the induction of its target genes. We would also note that the stress pathway induced by Top1 depletion is unusually restricted in comparison to many other stress responses that are often grouped together as the Environmental Stress Response (ESR; *Gasch et al., 2000*), since Msn2/4 target genes are not induced and RiBi genes are not downregulated (*Figure 1A,G*).

## Top1 depletion arrests ribosome biogenesis and activates a ribosomal assembly stress response

Although it may seem surprising that depletion of topoisomerases can induce a Hsf1-dependent stress response, formation of distinct nuclear foci by the Btn2 aggregase and perinuclear accumulation of the proteasome subunit Pre6 following Top1/2 degradation (*Figure 2—figure supplement 1A,B*) both point to the induction of proteotoxic stress in the nucleus (*Miller et al., 2015a*). Top1 was initially identified through a mutation (*mak1*) defective in large ribosomal subunit production (*Thrash et al., 1985*) and was later shown to be required for proper rRNA synthesis (*Brill et al., 1987*; *El Hage et al., 2010*; *French et al., 2011*). Consistent with these findings, we observed a strong reduction of pre-rRNA synthesis as shown by decreased [$^3$H]-adenine pulse-labeling of the RNAPI-transcribed 35S pre-rRNA, and two co-transcriptionally cleaved products, 27S and 20S pre-rRNAs, as early as 10 min after initiation of Top1 (or Top1 and Top2) depletion by addition of auxin to the medium (*Figure 2—figure supplement 1C*). This decreased rRNA synthesis is accompanied by an elongation defect, as shown by the accumulation of truncated pre-rRNAs that were initially described by the Tollervey laboratory (*El Hage et al., 2010*; *Figure 2A*). Further downstream, the rapid defect in rRNA production caused by inhibition of RNAPI elongation leads to unbalanced production of 40S and 60S ribosomal subunits, with a marked deficiency of the large (60S) subunit relative to the small (40S) subunit (*Figure 2B*). This would be expected to create a disequilibrium between RP and rRNA production, and more specifically an excess of unassembled RPs. Consistent with this, we detect accumulation of both and large and small subunit proteins (Rpl3 and Rps8, respectively) in trailing fractions of polysome gradients (*Figure 2C*). These observations strongly suggest that RPs fail to be incorporated normally into ribosomes immediately following topoisomerase degradation. In addition, this sedimentation profile may also reflect the presence of disassembling or incompletely assembled pre-60S particles. RPs are known to be prone to aggregation (*David et al., 2010*; *Pillet et al., 2017*; *Rand and Grant, 2006*; *Weids et al., 2016*) and recent reports show that newly synthetized, unassembled RPs accumulate in aggregates in response to ribosome assembly stress (*Sung et al., 2016a*; *Sung et al., 2016b*). Significantly, we also observed accumulation of RPs in an insoluble fraction following topoisomerase degradation (*Figure 2D*).

The observations described above led us to hypothesize that the transcriptional response to Top1 degradation is a consequence of defective ribosome assembly, perhaps driven by the proteotoxic stress caused by the accumulation of unassembled RPs. To challenge this idea, we measured the transcriptional response to three different perturbations to ribosome biogenesis: depletion of two essential ribosome assembly factors (Utp8 and Utp13) and treatment of cells with diazaborine. Utp8p is a member of the t-UTP subcomplex of 90S pre-ribosomes and its depletion inhibits rDNA transcription, leading to a reduction of the primary 35S pre-rRNA transcript and subsequent processing intermediates (*Gallagher et al., 2004*). In contrast, depletion of Utp13 (a member of the UTP-B subcomplex) interferes with downstream processing and synthesis of 40S subunits and causes decreased 18S rRNA levels without affecting the levels of the 25S or 5.8S rRNAs (*Gallagher et al., 2004*). Diazaborine, an inhibitor of the essential Drg1 AAA-ATPase, rapidly blocks mid-late steps of 60S subunit maturation (*Loibl et al., 2014*). Remarkably, all these treatments triggered a similar transcriptional response to that which occurs following Top1 depletion, namely a specific downregulation of RP genes and upregulation of Hsf1 target genes (*Figure 2E–G*; *Supplementary file 2*), which we refer to as the 'Ribosome Assembly STress Response' (RASTR).

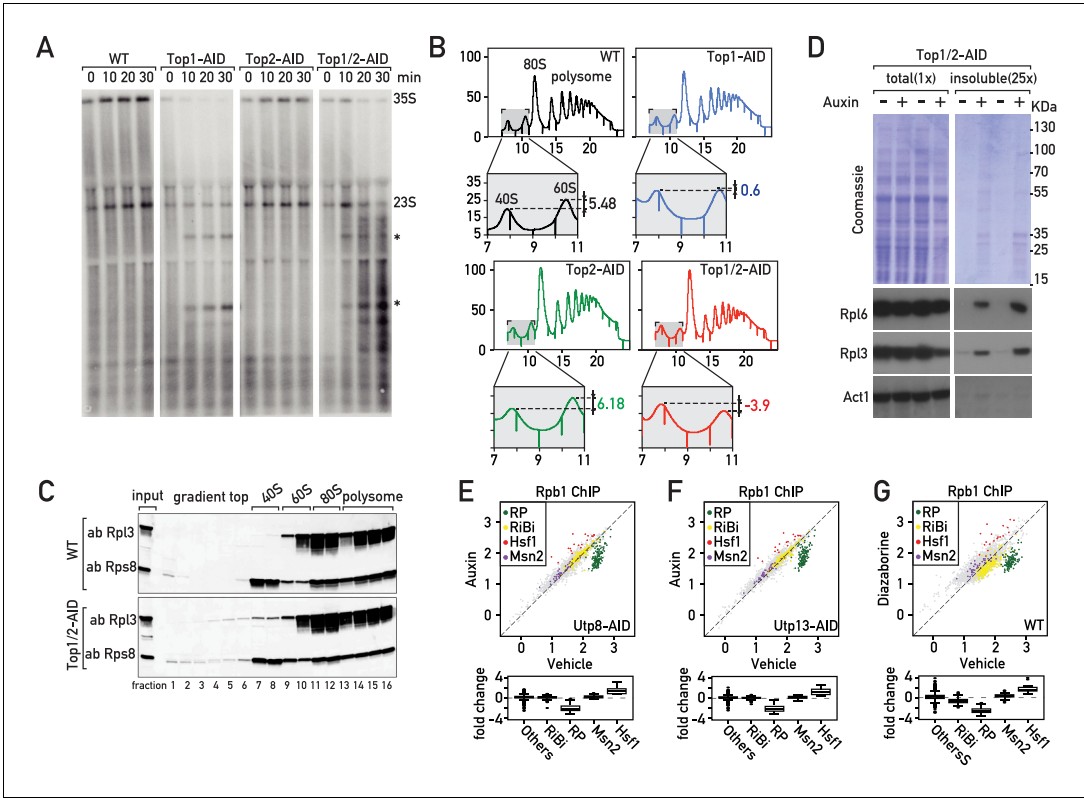

**Figure 2.** The effects of Top1 depletion on RNAPII regulation are linked to an underlying defect in ribosome biogenesis. (**A**) Northern blot of 5'ETS1-containing rRNAs prepared from cultures of wild-type, Top1-AID, Top2-AID and Top1/2-AID cells that had been pulse-labeled for 2 min with [³H] adenine at the indicated times following addition of auxin to the media. Total RNAs were extracted and samples were separated on agarose gels, transferred to a nylon membrane and first directly autoradiographed to reveal pulse labeling of nascent rRNAs (see **Figure 2—figure supplement 1C**). The membrane was next hybridized with a ³²P-labeled oligonucleotide probe allowing detection of all species containing 5'ETS1 (ACGACAAGCCT-ACTCGAATTCGT). Truncated pre-rRNA fragments, first identified in cells lacking Top1 (**El Hage et al., 2010**), are indicated (*). (**B**) Polysome sedimentation profiles (OD₂₆₀) of WT, Top1-AID, Top2-AID, and Top1/2-AID strains 20 min following auxin treatment (large panels, as indicated). The top of each gradient (fractions 7 to 11), corresponding to 40S and 60S subunit peaks, is expanded below, where peak height differences (60S:40S ratio) are indicated. (**C**) Total cell extracts prepared from the indicated fractions of sedimentation profiles of WT and Top1/2-AID strains (from B) were TCA precipitated and analyzed by Western blot following SDS-PAGE, using an antibody against Rpl3 and Rps8, as indicated. (**D**) Total (left panels) and detergent-insoluble pellet (right panels) fractions isolated from lysates of Top1/2AID cells treated (+) or not (-) with auxin were analyzed by SDS-PAGE and Coomassie blue staining (top panels) or immunoblotting with the indicated antibodies (bottom panels). The pellet fraction is overloaded 25-fold compared to the total extract fractions. (**E, F, G**) Scatter plots (top panels) comparing RNAPII (Rpb1) ChIP-Seq read counts for individual genes in Utp8-AID (**E**) or Utp13-AID (**F**) cells after 20 min of auxin or vehicle treatment, or WT cells after 20 min of treatment with diazaborine or vehicle (**G**) (y-axis: auxin or diazaborine) for 20 min versus non-depleted cells (x-axis, Vehicle). Each dot represents a gene (5041 genes in total) and genes are color-coded according to functional groups, as in **Figure 1A**. Bottom panels display the corresponding box plots for the four indicated gene categories plus all other genes (others).

DOI: https://doi.org/10.7554/eLife.45002.005

The following figure supplement is available for figure 2:

**Figure supplement 1.** Protein localization and transcriptional effecs of Top1, Top2 and Top1/2 depletion.

DOI: https://doi.org/10.7554/eLife.45002.006

Hsf1 activity is stimulated by many different types of cellular stress, including stalled ribosomes. A pioneering study reported that a set of proteins termed the RQC binds to 60S ribosomal subunits containing stalled polypeptides and leads to their degradation. In the process, the RQC triggers a specific stress signal that leads to Hsf1 target gene activation (**Brandman et al., 2012**). Thus, cells

lacking a component of the RQC, the Tae2 protein, fail to activate Hsf1 following translational stress. To ask if RASTR might be related to the RQC, we induced Top1/2 degradation in *tae2-Δ* cells. We found that activation of two Hsf1 target genes (*SSA1* and *HSP42*) and downregulation of two RP genes (*RPL30* and *RPL39*) was unaffected by deletion of *TAE2* (*Figure 2—figure supplement 1D*) and conclude that RQC does not play a role in RASTR. These results highlight that cells have developed distinct mechanisms to adapt the Hsf1 transcriptional program to defects in both ribosome activity and ribosome assembly.

## Ifh1 sequestration in an insoluble nuclear fraction during RASTR is driven by RP accumulation

Although many studies would support the notion that Hsf1 activation during RASTR occurs through sequestration of its inhibitory partner Hsp70 by RP aggregates (*Krakowiak et al., 2018*; *Shi et al., 1998*; *Zheng et al., 2016*), it is less clear how ribosome assembly stress could trigger release of Ifh1 from RP gene promoters. We reported previously that the association of Ifh1 with RP gene promoters in growing cells is rapidly disrupted (within 5 min) following inhibition of the growth-promoting TORC1 kinase by addition of rapamycin to the medium (*Schawalder et al., 2004*). More recently (*Albert et al., 2016*), we found that stable release of Ifh1 from RP gene promoters (measured 20 min after rapamycin addition) requires its C-terminal domain together with a complex of proteins containing casein kinase 2 (CK2) and two RiBi factors, Utp22 and Rrp7, with which Ifh1 interacts to form the CURI complex (*Rudra et al., 2007*). Thus, in *ifh1-ΔC* cells the truncated protein is rapidly released but later returns to RP gene promoters following TORC1 inhibition. This led us to propose two distinct mechanisms controlling the promoter release of Ifh1 following stress: one operating at a short timescale (<5 min) and the other on a long timescale (~20 min). Interestingly, ifh1-ΔC promoter release is stable following Top1 depletion, suggesting that an unknown mechanism regulates Ifh1 during RASTR (*Figure 3A*).

The fact that RP gene repression and Hsf1 target gene activation occur with identical kinetics following RASTR activation (*Figure 3B*), and that Ifh1 concentrates in nuclear foci rapidly after topoisomerase depletion (*Figure 3C*), suggests that Ifh1 could be sensitive to the accumulation of unassembled RPs in the nucleus, as is presumably the case for Hsf1. Several lines of evidence are consistent with this hypothesis. To begin with, in cells lacking Tom1, an E3 ligase required for degradation of unassembled RPs (*Sung et al., 2016a*; *Sung et al., 2016b*), but not in *TOM1* cells, Ifh1 accumulates in prominent nuclear foci even in the absence of stress (*Figure 3D*). This suggests that Ifh1 aggregates in cells that are unable to efficiently degrade excess RPs, even under optimal growth conditions. Consistent with this, the published mass spectrometry data of insoluble fractions from cells either treated with the proteasome inhibitor bortezomib or lacking Tom1 clearly identified Ifh1, together with RPs, RiBi proteins and two Hsp70 proteins, Ssa1 and Ssa2, inhibitory partners of Hsf1 (*Figure 3—figure supplement 1A*). These data indicate that Ifh1 could be trapped in an insoluble cellular fraction in the absence of Tom1 and thus decrease the pool of Ifh1 able to bind with RP gene promoters. To test this possibility, we combined deletion of *TOM1* with a mutant allele of *IFH1* (*ifh1-AA*) that weakens its interaction with RP gene promoters. Remarkably, *tom1-Δ* is synthetically lethal with *ifh1-AA* (*Figure 3E*) supporting the notion that RP aggregation could directly impact on Ifh1 promoter binding. Lastly, to exclude the possibility that the genetic interaction between *TOM1* and the mutated allele of *IFH1* could be linked to the growth defect of this mutation, we examined another mutated allele of IFH1 (*ifh1-6*) that triggers a similar growth defect (*Figure 3E*). Importantly, we showed in a previous study that this *ifh1-6* mutant protein remains bound at high levels to RP genes promoter even under stress conditions (*Albert et al., 2016*). Remarkably, *tom1-Δ* is not synthetically lethal with *ifh1-6* (*Figure 3E*), supporting the notion that genetic interaction with *ifh1-AA* is directly linked to the ability of Ifh1 to bind RP gene promoters.

To assess directly whether Ifh1 is sequestered in aggregates during RASTR, we analyzed by mass spectrometry the insoluble fraction following topoisomerase depletion. As previously reported for *tom1-Δ* cells (*Sung et al., 2016a*), the insoluble fraction is enriched in chaperones and RPs (*Figure 3F,G*). We also noted a strong increase in RiBi factors, primarily those implicated in biogenesis of the large ribosomal subunit (*Figure 3G*). Importantly, Ifh1 was never detected in an insoluble fraction in the absence of stress but was invariably detected in these fractions following topoisomerase depletion (*Supplementary file 3*). This rapid sequestration of Ifh1 may be sufficient to explain the observed downregulation of RP genes during RASTR.

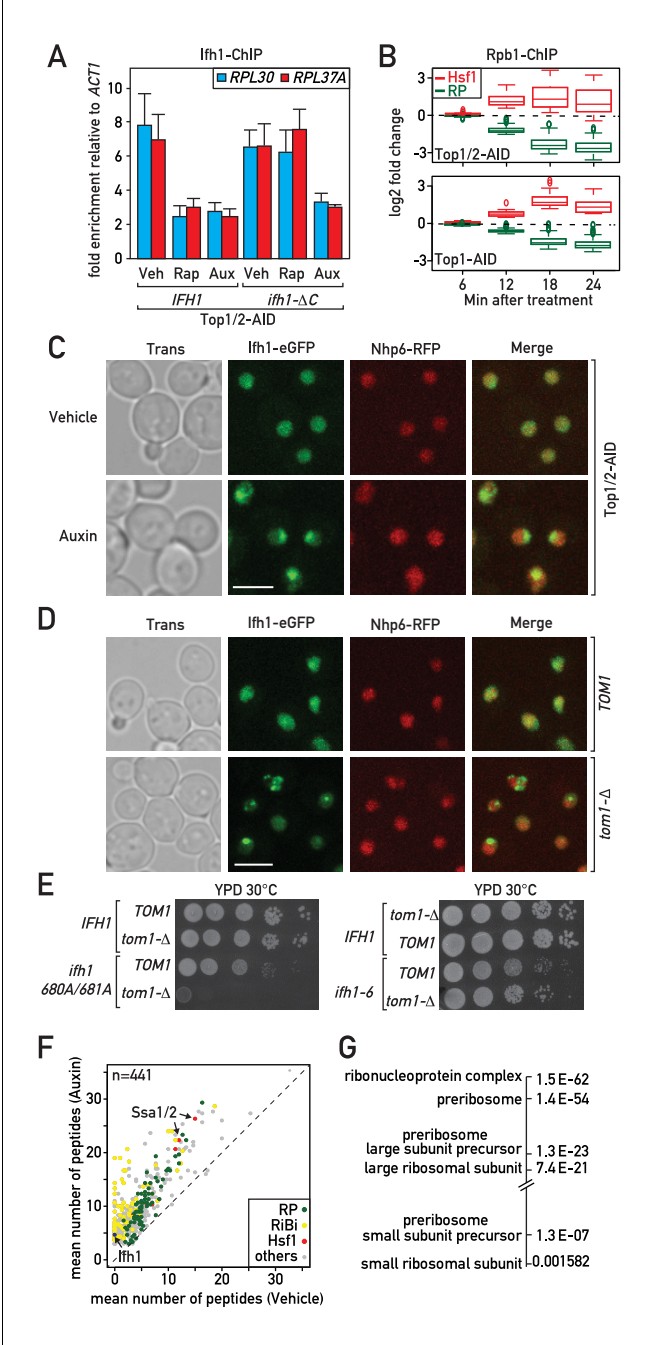

**Figure 3.** Evidence that Ifh1 is rapidly removed from RP gene promoters and sequestered in an insoluble nucleolar fraction pursuant to unassembled RP accumulation following RASTR initiation. (**A**) Ifh1 occupancy at the *RPL30* and *RPL37A* promoters 20 min following vehicle, rapamycin or auxin treatment of Top1/2-AID strains expressing either WT Ifh1 (*IFH1*) or a C-terminal truncated allele (*ifh1-ΔC*). Bar height indicates the average and error bars the range of N = 4 biological replicates. (**B**) Box plots showing the kinetics of RNAPII ChIP-seq changes at Hsf1 target and RP genes at the indicated time points (minutes) following auxin treatment in Top1/2-AID (top panel) or Top1-AID (bottom panel) strains. (**C**) A Top1/2-AID strain expressing Ifh1-eGFP and Nhp6-mCherry was grown exponentially and cell samples were used for fluorescence microscopy analysis after 20 min of auxin (Aux) or vehicle (Veh) treatment. (**D**) *TOM1* (top panels) and *tom1-Δ* (bottom panels) strains expressing Ifh1-eGFP and Nhp6-mCherry were grown exponentially and cell samples were used for fluorescence or transmission microscopy analysis, as indicated. (**E**) Tenfold serial dilutions of *IFH1*, *ifh1-AA* (*ifh1 680A/681A*) or *ifh1-6* cells transformed in either *TOM1* or *tom1-Δ* backgrounds (as indicated) were grown in YPD medium for 44 hr at 30°C before being photographed. (**F**) Scatter plot comparing average number of peptides purified in an insoluble fraction from Top1/

*Figure 3 continued on next page*

*Figure 3 continued*

2-AID cells treated for 20 min with either auxin (y-axis, Auxin) or vehicle (x-axis, Vehicle). Each dot represents a protein, color-coded according to functional group as above (green: RP, red: Hsf1 target gene product, yellow: RiBi protein, gray: others), with some specific proteins indicated by arrows. (G) Gene Ontology and p-values of protein groups that are the most enriched in the insoluble fraction following Top1/2-AID depletion (Δ > 3 peptides in insoluble fraction after topoisomerase depletion compared to vehicle in all experiments, n = 3 biological replicates).

DOI: https://doi.org/10.7554/eLife.45002.007
The following figure supplement is available for figure 3:

**Figure supplement 1.** Protein localization and transcriptional effecs of Top1, Top2 and Top1/2 depletion.
DOI: https://doi.org/10.7554/eLife.45002.008

## Neo-synthetized RPs are required for RASTR activation

Given their fast turnover rate, nuclear accumulation in the absence of ribosome assembly and propensity to aggregate, unassembled RPs could be ideally positioned to rapidly signal ribosome biogenesis defects (*Sung et al., 2016a*; *Milkereit et al., 2001*). To evaluate the importance of newly synthetized RPs in RASTR, we blocked their production by cytoplasmic anchoring of Ifh1 before topoisomerase depletion (*Figure 4A*). It is important to note that Ifh1 binding is highly specific to RP genes (*Knight et al., 2014*) and that the transcriptional effect of its nuclear depletion is restricted to RP genes and a very small number of additional targets (*Supplementary file 4*). Although Ifh1 depletion by anchor-away may not be complete (the strain still grows on plates containing rapamycin, albeit slowly, even though Ifh1 is essential for growth; *Figure 4—figure supplement 1A*), it

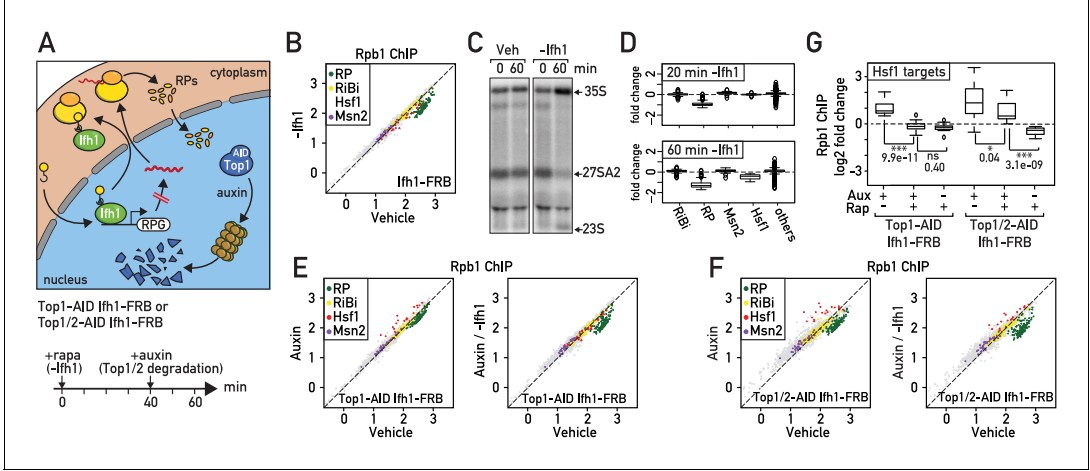

**Figure 4.** Downregulation of RP gene expression by Ifh1 nuclear depletion prior to RASTR initiation strongly dampens Hsf1 target gene activation. (A) Schematic of protocol for Ifh1-FRB nuclear depletion (0–60 min of rapamycin treatment) followed by Top1-AID or Top1/2-AID degradation (auxin treatment, 40–60 min). (B) Scatter plot comparing RNAPII (Rpb1) ChIP-seq in Ifh1-FRB cells either rapamycin-treated for 60 min (y-axis, -Ifh1, Ifh1-FRB nuclear depletion) or untreated (x-axis, Vehicle, no Ifh1-FRB depletion). Categorization and color coding of genes as above. (C) Northern blots of pre-rRNA after 0 or 60 min of Ifh1-FRB nuclear depletion by anchor-away (-Ifh1) or vehicle (Veh) treatment of Ifh1-FRB strain. (D) Box plots of the data shown in (B) for the indicated five gene categories, showing fold-change upon Ifh1-FRB nuclear depletion compared to mock-treated cells. (E) Scatter plots comparing RNAPII (Rpb1) ChIP-seq in Top1-AID Ifh1-FRB cells either auxin-treated (y-axis, Aux, left panel) or auxin- plus rapamycin-treated (y-axis, Aux / -Ifh1, right panel) treated, as described in (A), versus untreated cells (x-axis, vehicle, both panels). (F) As in (E), but for Top1/2-AID Ifh1-FRB cells. (G) Box plots showing RNAPII (Rpb1) ChIP-seq change after rapamycin and/or auxin treatment for Hsf1 target genes in Top1-AID Ifh1-FRB cells (left) or Top1/2-AID Ifh1-FRB cells (right). Asterisks show significant difference according to student's t-test (*: p<0.05, ***: p<0.001, ns: Not significant), p-value is indicated.

DOI: https://doi.org/10.7554/eLife.45002.009
The following figure supplement is available for figure 4:

**Figure supplement 1.** Characterization of Ifh1-FRB Top1-AID Top2-AID and Ifh1-FRB Top1-AID strains, and binding of Ifh1at Hsf1 target and RP gene promoters.
DOI: https://doi.org/10.7554/eLife.45002.010

nonetheless leads to a significant and highly specific decrease in RP gene transcription as measured by RNAPII ChIP-seq (*Figure 4B*; *Supplementary file 4*). Interestingly, Ifh1 depletion also leads to aberrant rRNA processing (*Figure 4C*) as would expected in conditions where RP levels become limiting (*Reiter et al., 2011*). Remarkably, we noted that 60 min of Ifh1 anchoring alone, in the absence of topoisomerase depletion, also caused a significant down-regulation of Hsf1 target genes (*Figure 4D*; *Supplementary file 4*) even though Ifh1 is absent from the promoters of these genes (*Figure 4—figure supplement 1B*), suggesting that the Hsf1 transcriptional program is continuously influenced by RP production. Consistent with this idea, we found that upregulation of Hsf1 target genes was either abolished or strongly reduced (*Figure 4E,F,G*; *Supplementary file 4*) when Top1 or Top1 and Top2 were degraded following nuclear depletion of Ifh1, indicating that RP production is required for Hsf1 target gene activation during RASTR.

## Cells balance RP production and Hsf1 activity even in the absence of stress

As an alternative approach to test the requirement for de novo RP synthesis to initiate RASTR we used cycloheximide treatment, which blocks all translational elongations, thus leading to rapid depletion of the nuclear pools of RPs (*Figure 5A*; *Gorenstein and Warner, 1977*; *Warner, 1977*; *Reiter et al., 2011*; *Lam et al., 2007*). As reported by others (*Reiter et al., 2011*), we confirmed that cycloheximide alone also triggers a rapid arrest of rRNA processing (*Figure 5B*). Quite strikingly, we observed a transcriptional response to cycloheximide treatment exactly opposite to that induced by RASTR, namely Hsf1 target gene downregulation and RP gene upregulation (*Figure 5C*; *Supplementary file 5*). This finding suggests that even in unstressed cells RP production may contribute to a basal level of Hsf1 activation while at the same time limiting Ifh1 activity at RP gene promoters.

Significantly, treatment of cells undergoing Top1 or Top1/2 depletion with cycloheximide (auxin + CHX) completely abolished both RP gene repression and activation of Hsf1 target genes (*Figure 5D,E*; *Supplementary file 5*). Indeed, Hsf1 target gene activation under these conditions is lower than in untreated cells and not significantly different than that seen in cells treated with cycloheximide alone (*Figure 5—figure supplement 1A*). These findings clearly demonstrate that RASTR is dependent upon de novo protein synthesis. Importantly, it was recently reported that cycloheximide treatment efficiently prevents aggregation of newly synthesized RPs following proteasome inhibition (*Sung et al., 2016a*). Perhaps as a direct consequence of this, we found that CHX treatment also leads to strong reduction of Ifh1-eGFP nuclear foci that are observed in cells lacking the ubiquitin ligase Tom1, which is specifically required for efficient degradation of unassembled RPs (*Sung et al., 2016a*; *Sung et al., 2016b*; *Figure 5—figure supplement 1B*). Furthermore, Ifh1 disaggregation following cycloheximide exposure is associated with increased Ifh1 binding at a RP gene promoter, which becomes significant in *tom1-Δ* cells (*Figure 5—figure supplement 1C*). Similarly, cycloheximide treatment also prevents the release of Ifh1 from RP gene promoters in response to the activation of RASTR by topoisomerase depletion (*Figure 5—figure supplement 1D*). Considered as a whole, these data suggest that both RP and Ifh1 subnuclear structures (aggregates) are dynamic, promoted by de novo RP production upon RASTR initiation, and capable of influencing Ifh1 promoter binding. Consistent with this view, we observed a large increase in cells that accumulate RP (Rpl25) or Ifh1 nuclear aggregates during RASTR that is abolished in the presence of cycloheximide (*Figure 5F,G*; *Figure 5—figure supplement 1E, F*).

Taken together with the strong reduction of Hsf1 target gene activation following Ifh1 cytoplasmic anchoring, both in the presence and absence of topoisomerase degradation, our observations on the effect of cycloheximide highlight the interwoven nature of RP and Hsf1 target gene regulons and support the notion that unassembled, aggregated RPs constitute the primary RASTR-induced signal capable of regulating both Ifh1 and Hsf1 activities, albeit in an opposite direction. More generally, these data indicate that newly synthetized RPs, in both stressed and unstressed cells, operate as a central hub in coordinating the expression of RP genes themselves with the Hsf1-dependent activation of chaperone and proteasome genes.

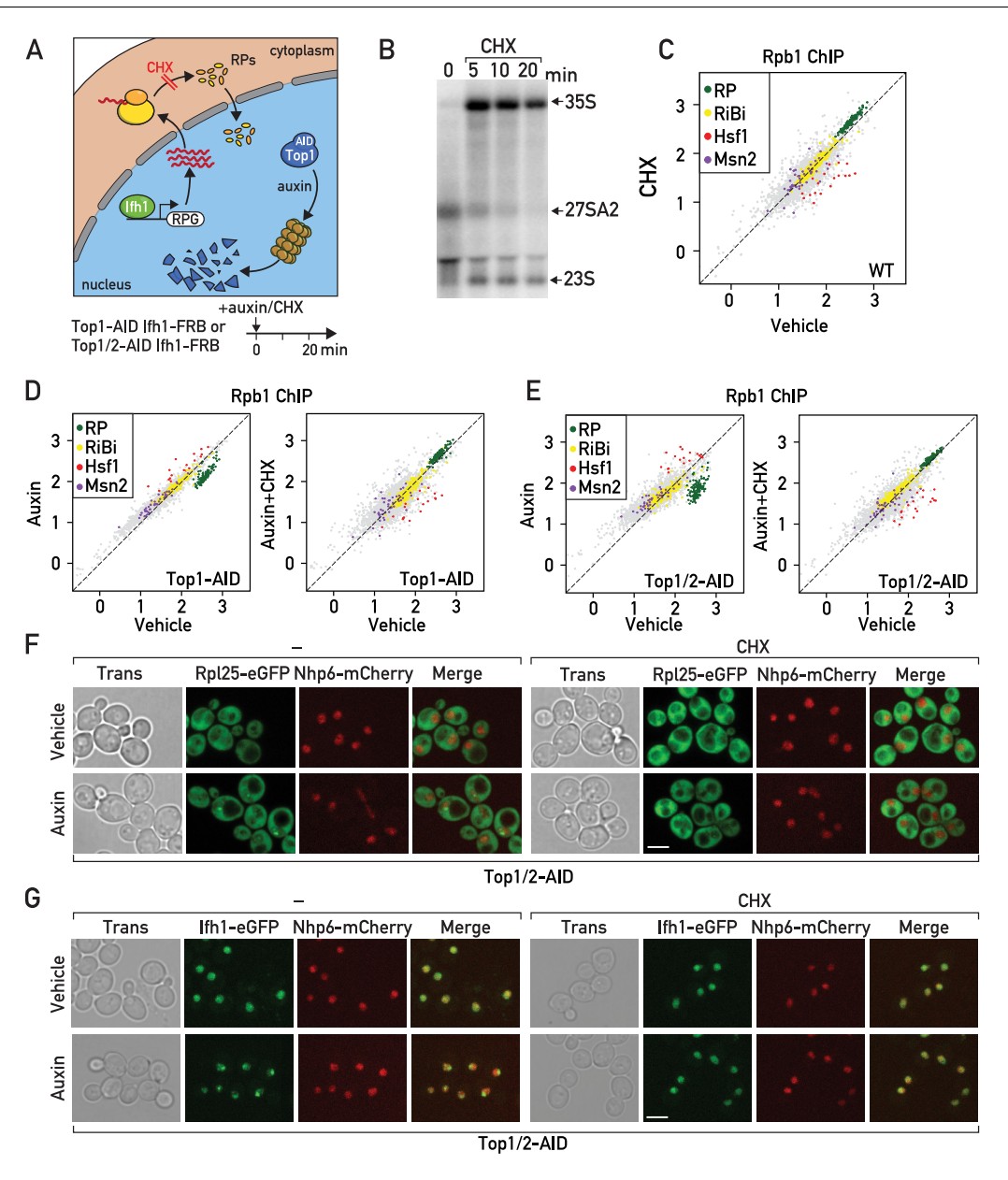

**Figure 5.** Cycloheximide treatment induces a rapid RNAPII transcriptional response opposite and epistatic to that of RASTR. (**A**) Schematic related to experiments in subsequent panels describing the effect of cycloheximide treatment on de novo RP production and auxin treatment on Top1-AID (or Top1/2-AID) degradation. (**B**) Northern blots of pre-rRNA after 0, 5, 10, and 20 min of cycloheximide (CHX) treatment. (**C**) Scatter plot comparing RNAPII (Rpb1) ChIP-seq after 20 min of cycloheximide treatment (y-axis, CHX) to that of non-treated cells (x-axis, vehicle) at the indicated groups of target genes. (**D**) Scatter plots comparing RNAPII ChIP-seq in auxin-treated to untreated cells (left panel) and in auxin + cycloheximide (CHX)-treated to untreated cells (right panel). In both cases cells express Top1-AID. (**E**) Scatter plots comparing RNAPII ChIP-seq as in (**D**), but for Top1/2-AID cells. (**F**) Box plots showing RNAPII ChIP-seq fold-change for Hsf1 target genes after cycloheximide (CHX) and/or auxin (Aux) treatment of Top1-AID or Top1/2-AID cells, as indicated (data taken from experiments shown in **D** and **E**). p-Values are shown above the indicated comparisons together with significance according to student's t-test (***: p<0.001, ns: not significant). (**G–H**) Top1/2-AID strains expressing Rpl25-eGFP (**G**) or Ifh1-eGFP (**H**) and Nhp6-mCherry were grown exponentially and samples were used for fluorescence microscopy analysis after 20 min of auxin (top) or vehicle treatment (bottom), in the absence (left) or presence (right) of cycloheximide (CHX).
DOI: https://doi.org/10.7554/eLife.45002.011

*Figure 5 continued on next page*

*Figure 5 continued*
The following figure supplement is available for figure 5:

**Figure supplement 1.** Characterization of Ifh1-FRB Top1-AID Top2-AID and Ifh1-FRB Top1-AID strains; binding of Ifh1 at Hsf1 target and RP gene promoters.
DOI: https://doi.org/10.7554/eLife.45002.012

## RASTR is the first transcriptional response to environmental stress

We next turned our attention to the potential involvement of RASTR during more general stress responses that might also rapidly affect ribosome assembly. In an initial set of experiments, we inactivated the conserved growth-promoting TORC1 kinase by treatment of cells with rapamycin, which is known to mimic a major part of the environmental stress response, including osmotic and redox stress, as well as carbon, nitrogen, phosphate or amino acid starvation (*Loewith and Hall, 2011*). As reported previously, rapamycin triggers a rapid arrest of rRNA processing (*Figure 6A*) and a decrease of RP and RiBi gene expression (*Figure 6B,C*; *Supplementary file 6*). Interestingly, we noted that Hsf1 target genes are transiently up- and downregulated at 5 and 20 min, respectively, following rapamycin addition (*Figure 6B,C*; *Supplementary file 6*), suggesting that RASTR is activated at the early time point but shortly thereafter turned off. Consistent with this view, it has been reported that RP production ceases around 15 min after rapamycin treatment (*Reiter et al., 2011*), which we suggest would turn off the signal for RASTR, thus explaining the downregulation of Hsf1 target genes observed at 20 min.

To explore this hypothesis further, we took advantage of our observation that cycloheximide treatment prevents RASTR activation by either Top1 or Top1/2 degradation (*Figure 5*) and treated cells with cycloheximide 5 min before rapamycin addition of either 5 or 20 min (*Figure 6D*, see schematics of experimental protocols below the respective panels). Remarkably, this specifically prevented RP gene repression and Hsf1 target gene activation at 5 min following rapamycin addition, whereas at the longer time point (20 min) RP genes and Hsf1 were regulated independently of cycloheximide (*Figure 6D,E*; *Supplementary file 6*). Consistent with the early block in RP gene downregulation being due to a failure to initiate RASTR immediately following rapamycin addition, we showed that cycloheximide pre-treatment prevents release of Ifh1 from RP gene promoters at 5 min, but not at 20 min following rapamycin treatment (*Figure 6F*).

The effects of rapamycin treatment described above are fully consistent with our previous report demonstrating that regulation of RP gene transcription following TORC1 inactivation by rapamycin operates through two distinct mechanisms at short and long timescales, with the latter dependent on RNAPI activity and the CURI complex (*Albert et al., 2016*). The short timescale mechanism described here, which is dependent upon continued protein synthesis and presumably mediated by RASTR, allows cells to rapidly arrest RP production and avoid or minimize proteotoxic stress induced by arrested ribosome assembly. The second mechanism permits the resumption of RP production only when rRNA synthesis also resumes (*Albert et al., 2016*). These two mechanisms could be particularly useful to rapidly adapt ribosome production to new growth conditions.

To explore the possible generality of rapid RASTR-mediated shut-down of RP gene transcription in response to stress, we measured the transcriptional response to heat shock, which is known to transiently downregulate both RP gene and rRNA transcription (*Gasch et al., 2000*; *Kos-Braun et al., 2017*). As expected, we observed strong downregulation of RP genes and upregulation of Hsf1 target genes only 5 min following a shift in temperature to 40°C (*Figure 7A–C*, left panels) that was also accompanied by strong upregulation of Msn2 target genes and downregulation of RiBi genes (*Figure 7A*, left panel). The heat-shock transcriptional response is thus much broader than that to ribosome assembly stress, despite their common effects on RP and Hsf1 target gene expression. Remarkably, though, we found that cycloheximide pre-treatment prevents the strong and immediate repression of RP genes following heat shock (*Figure 7A,B*; right panels; *Supplementary file 6*), consistent again with the idea that this facet of the heat-shock response is identical to that which occurs during RASTR. Importantly though, Hsf1 target genes are still activated following heat shock in the presence of cycloheximide (*Figure 7A,C*; right panels), presumably because the unfolding of thermo-labile proteins induced by heat shock is alone sufficient to activate Hsf1 even in the absence of continuing RP synthesis. Nevertheless, the striking requirement for de

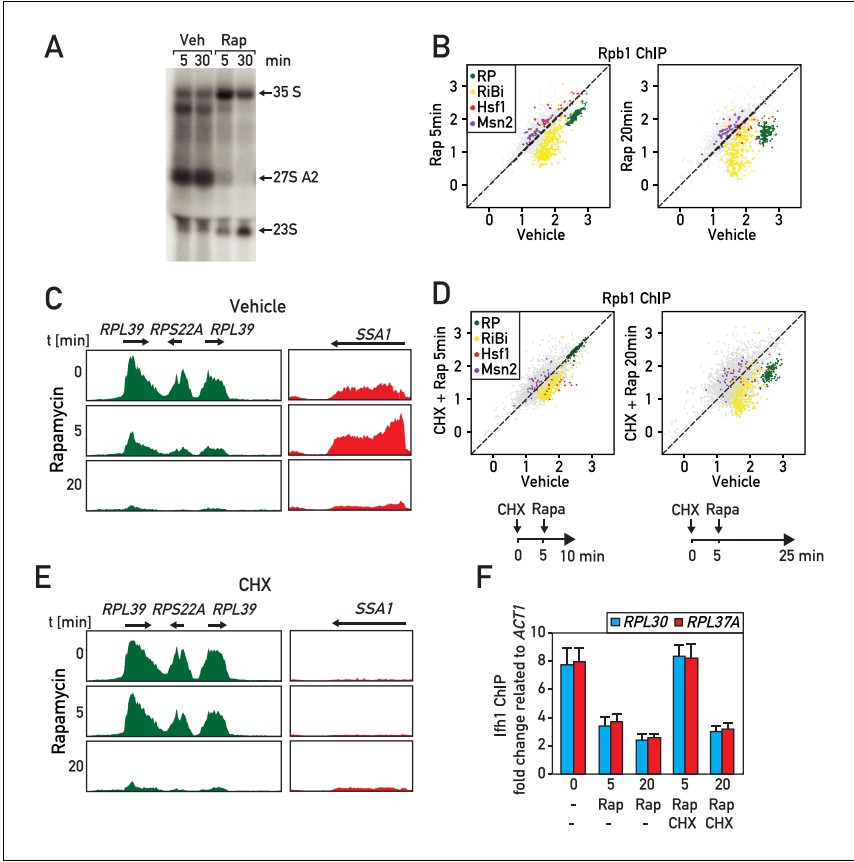

**Figure 6.** RASTR comprises the cycloheximide-sensitive component of the early RNAPII response to TORC1 inhibition. (**A**) Northern blots of pre-rRNA from wild type cells after 5 or 30 min of vehicle (Veh) or rapamycin (Rap) treatment. (**B**) Scatter plots comparing RNAPII (Rpb1) ChIP-seq in cells after 5 (y-axis, left panel) or 20 min (y-axis, right panel) of rapamycin (Rap) treatment to non-treated cells (x-axis, vehicle). Gene groups are color-coded as indicated. (**C**) Genome browser tracks showing RNAPII ChIP-seq read counts for three consecutive RP genes on chromosome X (in green, left panel) or at the *SSA1* gene on chromosome I (in red, right panel), following 0, 5, or 20 min (top, middle, bottom panels, respectively) of rapamycin treatment. Gene annotations (gene name, open reading frame and direction of transcription) are shown above the tracks. (**D**) Scatter plots comparing RNAPII ChIP-seq in cells pre-treated with cycloheximide (CHX) then treated for 5 (left panel) or 20 (right panel) minutes with rapamycin to cells untreated (x-axis, vehicle). Schematic representation of the experimental protocols is shown below each panel. Cells were collected for RNAPII ChIP-seq analysis after 5 (left panel) or 20 min (right panel) of rapamycin treatment. (**E**) Ifh1 occupancy at the *RPL30* and *RPL37A* promoters following 5 or 20 min of rapamycin treatment (Rap) in cells pre-treated or not with cycloheximide (CHX) for 5 min. Bar height indicates the average and error bars the range of N = 4 biological replicates.

DOI: https://doi.org/10.7554/eLife.45002.013

novo protein synthesis for RP gene downregulation following heat shock strongly suggests that RASTR plays an integral role in this component of the heat-shock response and thus may constitute the earliest transcriptional response, at the level of RNAPII, to a wide variety of stress conditions.

## Discussion

In this study, we demonstrate the existence of a regulatory mechanism, which we refer to as the ribosome assembly stress response, or RASTR, that allows yeast cells to specifically coordinate the activity of two TFs, Hsf1 and Ifh1, with the functional state of ribosome assembly. Our data and several previous reports suggest that rapid ribosome biogenesis is a potentially proteotoxic process, in large part due to accumulation of unassembled RPs, whose production needs to be carefully coordinated at the transcriptional level, at least in yeast, together with that of chaperones and proteasome

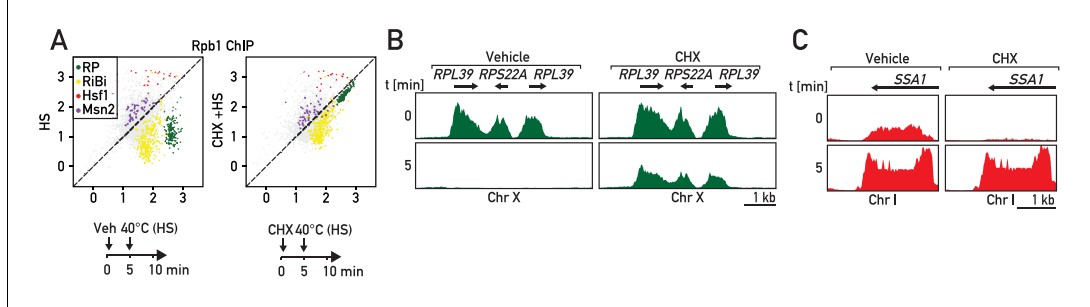

**Figure 7.** RP gene downregulation following heat shock is blocked by cycloheximide. (**A**) Scatter plots comparing RNAPII ChIP-Seq in cells pre-treated or not with cycloheximide (CHX) followed by 5 min of heat shock (y-axis, HS) versus non-stressed cells (x-axis, no HS, no CHX). Schematic of the protocol for each experiment is shown below the data panels: mock pre-treatment (vehicle; 0–5 min) followed by heat shock (HS; 40°C, 5–10 min; left panel) or cycloheximide (CHX) pre-treatment (0–5 min) followed by heat shock (HS; 40°C, 5–10 min; right panel). Samples for ChIP-seq analysis of RNAPII association were taken at 10 min in both experiments. (**B, C**) Genome browser tracks showing RNAPII ChIP-Seq read counts for the experiments described in (**A**) on a region on chromosome X containing three consecutive RP genes (in green, **B**) or at the *SSA1* gene on chromosome I (in red, **C**). Cells were either mock pre-treated (vehicle; left panels) or cycloheximide pre-treated (CHX; right panels) before heat shock (HS). Gene annotations (gene name, open-reading frame and direction of transcription) are shown above the tracks.
DOI: https://doi.org/10.7554/eLife.45002.014

components (*Figure 8*). In this perspective, RASTR may play an essential role in minimizing the proteostasis burden imposed by high ribosome production rates, particularly under fluctuating environmental conditions.

Our data indicate that disequilibrium at any step-in ribosome biogenesis (rRNA transcription, early or late rRNA processing or assembly) will lead to RASTR activation until the pool of unassembled RPs decreases. Consequently, proteasome inhibition (*Kos-Braun et al., 2017*), dNTP depletion (*Gómez-Herreros et al., 2013*), DNA damage (*Conconi et al., 2005*), nutrient and thermal stress (*Liu et al., 1996*; *Tsang et al., 2003*), all of which are known to alter rRNA transcription or processing, or ribosome assembly (reviewed in *James et al., 2014*), are likely to activate RASTR, as do the genetic perturbations at different step of ribosome assembly described here. Taken together with previous reports, our data thus point to a critical role for RASTR in the transcriptional networks regulating both growth and protein homeostasis.

It is important to note that RPs are among the most abundant ubiquitinylated proteins that accumulate in the nucleus of proteasome-deficient *S. cerevisiae* and human cells (*Sung et al., 2016a*; *Sung et al., 2016b*; *Lam et al., 2007*; *Mayor et al., 2007*), suggesting that the synthesis of RPs and their assembly into ribosomes must be tightly coordinated with the cell's proteostasis capacity. Consistent with this view, we show here that induction of ribosome assembly stress is correlated with the rapid accumulation of RPs in a detergent-insoluble fraction and that blocking de novo RP production, either by anchoring away Ifh1 or treating cells with cycloheximide, diminishes or abolishes a key transcriptional consequence of RASTR, namely upregulation of Hsf1 target genes. Our observations thus strongly suggest that RP aggregates are an important activating signal for RASTR. Nevertheless, we and others (*Sung et al., 2016a*) detect a large number of additional proteins that accumulate in an insoluble fraction upon ribosome assembly stress, including many RiBi proteins (e.g. numerous rRNA helicases and processing factors), the RP gene activator Ifh1, and chaperones. At present, we do not know the precise molecular nature of these aggregates, which presumably accumulate in the nucleolar space, or even whether they represent a common structure. In any event, our data suggest that RP- and Ifh1-containing aggregates are highly dynamic since the promoter release of Ifh1 and activation of Hsf1 following Top1 degradation are rapidly reversed by Top2 compensation (*Figure 1*) and Ifh1 aggregates that appear in *tom1-Δ* cells quickly diminish following cycloheximide treatment (*Figure 5—figure supplement 1A*). We imagine that this provides a strong selective advantage by allowing cells to rapidly recover from a transient disruption of ribosome biogenesis.

Accumulation of proteins in insoluble fractions or aggregates underlies numerous diseases, as well as aging (*Saarikangas and Barral, 2015*; *Tuite and Melki, 2007*). However, certain protein aggregates appear to be dynamic structures that contribute to cellular fitness by protecting the cell during stress (*Cherkasov et al., 2013*; *Douglas et al., 2008*; *Grousl et al., 2018*; *Kaganovich et al.,*

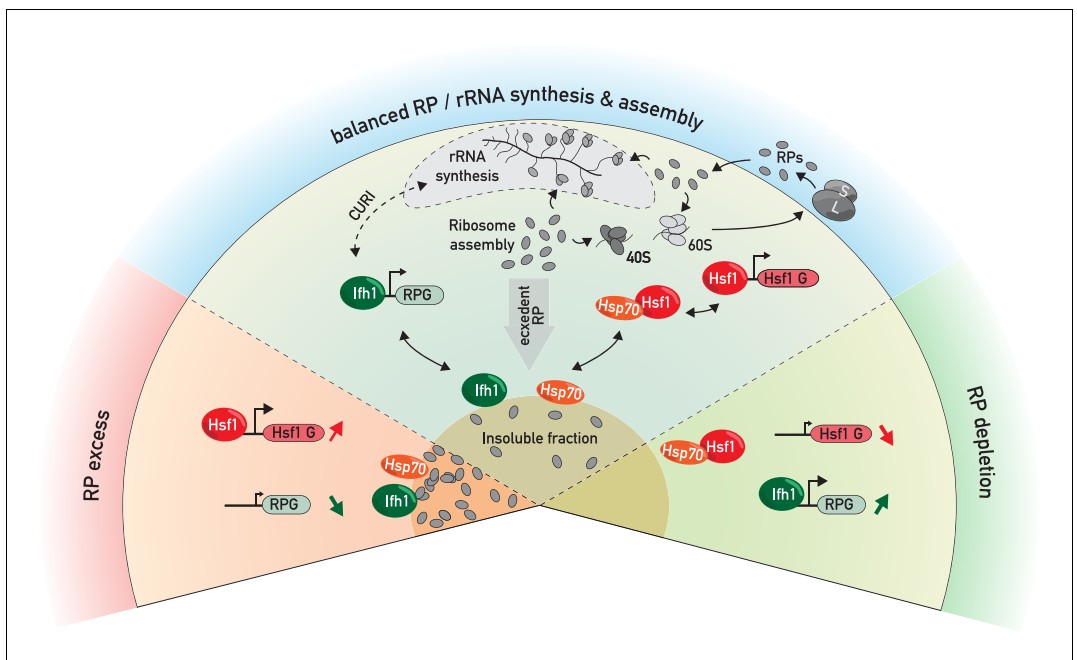

**Figure 8.** RP and Hsf1 target genes are regulated as a function of unassembled RP levels in both growing and stressed cells. In rapidly growing, un-stressed cells, RNAPII initiation at RP and Hsf1 target genes is continuously adjusted according to the levels of un-assembled RPs (central sector). Under various stress conditions (including RASTR, heat shock, TORC1 inhibition, perhaps many others), levels of unassembled RPs increase dramatically, and RPs accumulate with nucleolar proteins, Ifh1 and chaperones in an insoluble nuclear or nucleolar fraction. This leads to the rapid upregulation of Hsf1 target genes (e.g. chaperones and proteasome components), presumably through Hsp70 titration, and to the coincident downregulation of RP genes, through Ifh1 sequestration (bottom left sector, 'RP excess'). Conversely, a decrease in RP production (as provoked here by cycloheximide treatment) will lead to an opposite transcriptional response (bottom right sector, 'RP depletion'). In summary, we propose that levels of unassembled nuclear RPs act to constantly adjust RP and Hsf1 target gene expression, allowing the cell to balance growth with protein homeostasis.
DOI: https://doi.org/10.7554/eLife.45002.015

*2008*; *Miller et al., 2015b*). For example, stress granules and P-bodies, two of the most intensively studied insoluble macromolecular aggregates, have emerged as important cytoplasmic regulators of gene expression by controlling the processing, sequestering and/or degradation of specific RNA transcripts (*Decker and Parker, 2012*; *Mahboubi and Stochaj, 2017*). Interestingly, it has recently been reported that nucleolar proteins can form sub-compartmental structures by promoting liquid-liquid phase separation (*Berry et al., 2015*; *Feric et al., 2016*). Although further work will be required to characterize the composition, assembly and function of these nucleolar membrane-less structures, they are attractive candidates for regulatory hubs that could act by sensing ribosome bio-genesis stress and controlling adaptive responses. With respect to the present study, we imagine that liquid phase-separated structures in the nucleolus could be directly involved in sequestration of Ifh1 during RASTR, as well as the titration of Hsp70 that we propose leads to Hsf1 activation. A challenge for future studies will be to characterize the physical properties of these postulated structures and their relevance to the transcriptional outputs that we measure here.

We showed previously (*Albert et al., 2016*) that in strains where RNAPI is constitutively active (*Laferté et al., 2006*) Ifh1 is released from RP gene promoters shortly after TORC1 inhibition by rapamycin treatment (~5 min) but returns only 15 min later. This promoter re-binding does not occur in wild-type cells due to the action of two RiBi proteins (Utp22 and Rrp7) that can sequester Ifh1 in the CURI (Casein Kinase 2/Utp22/Rrp7/Ifh1) complex (*Albert et al., 2016*; *Rudra et al., 2007*) allow-ing to re-align RP gene expression with RNAPI activity. These findings revealed that *S. cerevisiae* has developed temporally distinct mechanisms to regulate RP gene expression. One of the keys finding in the present study is to confirm the existence of a two-step process in RP gene regulation and to

link the first step to control of the Hsf1 regulon. We propose that this short timescale mechanism results from a rapid rise in unassembled RPs that occurs immediately following TORC1 inhibition, or other stresses that disrupt ribosome assembly, such as depletion of Top1 or RiBi factors.

Following ribosome assembly stress, we imagine that the rapid induction of Hsf1 target genes, in combination with the arrest of RP gene transcription, contributes to the eventual clearing of proteotoxic unassembled RPs from the nucleus, thereby removing the signal that promotes RASTR and leads to Ifh1 promoter release and Hsf1 activation. Accordingly, RP production is abolished by 20 min following TORC1 inhibition (*Reiter et al., 2011*), suggesting that RASTR becomes inoperant, thus explaining the downregulation of Hsf1 genes (*Figure 6B*) and the switch to a secondary regulatory mechanism, involving sequestration of Ifh1 in the CURI complex, to align RP expression with rRNA production (*Albert et al., 2016*). Consistent with this view, the short time scale mechanism of RP gene downregulation is abolished by attenuating proteotoxicity through translation inhibition (cycloheximide treatment) whereas the long time scale process is insensitive to translation arrest but can be prevented by expression of a constitutively active RNAPI (*Albert et al., 2016*; *Laferté et al., 2006*). These two independent mechanisms adapt RP gene expression to both rRNA production and ribosome assembly, thus minimizing the accumulation of unassembled RPs.

As alluded to above, ribosome assembly stress in higher eukaryotes has been studied extensively in the context of ribosomopathies, diseases often associated with RP gene haplo-insufficiencies, RP gene point mutations or mutations in RiBi factors. One hypothesis put forward to explain these observations is that unassembled RPs trigger a feedback mechanism that decreases transcription of ribosome biogenesis genes by inhibiting c-Myc function and arrests cell growth through p53 activation (*Dai et al., 2007*; *Liu et al., 2016*). It has also been recently reported that the rRNA helicase DDX21 binds to and activates RP gene promoters in a manner that may be sensitive to the status of ribosome biogenesis (*Calo et al., 2015*). Taken together, these findings suggest that unassembled RPs could mediate an ancestral process to regulate ribosome biogenesis conserved from prokaryotes (*Nomura, 1999*) to eukaryotes. Transcriptome analysis immediately following ribosome assembly stress in mammalian cells will be required to understand the interplay between these different mechanisms and may also uncover novel pathways.

Our work also provides insights into the connection between ribosome assembly and Hsf1 that was first revealed in a report from the Churchman lab that appeared as our work was being prepared for publication (*Tye et al., 2018*). Hsf1 is a key sensor of proteotoxic stress in all eukaryotes that controls a common set of chaperones conserved from yeast to human. One protective function reported for Hsf1 is its ability to reduce protein aggregate formation leading to neurogenerative diseases (*Neef et al., 2014*). On the other hand, Hsf1 also exerts a pro-oncogenic function through its ability to promote proteostasis in rapidly growing tumor cells (*Mendillo et al., 2012*; *Santagata et al., 2011*). Despite its central function, a holistic understanding of the regulatory mechanisms that govern Hsf1 activity still missing. Our work and that of *Tye et al. (2018)* demonstrates that Hsf1 activity is tightly linked to ribosome biogenesis in yeast, in a manner independent of the previously described RQC mechanism that contributes to the dissociation of aberrant nascent polypeptides from the ribosome (*Brandman et al., 2012*). These two mechanisms highlight the central importance of ribosome assembly and activity in regulation of cellular protein homeostasis through Hsf1. Although it is currently unknown if RASTR is conserved in metazoans, we note that RPs are also subjected to a high turnover rate compared to other nuclear components in mammalian cells and that proteasome or ribosome assembly inhibition trigger a rapid accumulation of RPs in the nucleus, whereas arrest of translation has an opposite effect (*Sung et al., 2016a*; *Lam et al., 2007*). Importantly, it was reported that cycloheximide treatment also abolishes Hsf1 activity in mammalian cell by an unknown mechanism (*Santagata et al., 2013*). We propose that a dynamic balance between unassembled and assembled RPs could be sensed by Hsf1 to constantly adjust protein homeostasis transcription programs in eukaryotes with translational flux, proteolysis and the rate of ribosome assembly (*Figure 8*), since disruption or hyperactivation of any of these processes will rapidly change nuclear levels of free RPs. Given the growing body of evidence linking Hsf1 activity to numerous diseases associated with proteotoxic stress, but also rapid cell growth in cancer, it will be of great interest to challenge this model in the future.

# Materials and methods

## Key resources table

| Reagent type | Designation | Source or reference | Identifiers | Additional information |
|---|---|---|---|---|
| Chemical compound, drug | Cycloheximide | Sigma | C7698 | Materials and methods subsection: Yeast strains and growth |
| Chemical compound, drug | Rapamycin | Sigma | R8781 | Materials and methods subsection: Yeast strains and growth |
| Chemical compound, drug | Auxin | Sigma | 1288G | Materials and methods subsection: Yeast strains and growth |
| Chemical compound, drug | Diazaborine | Provided by H Bergler. Zisser et al., 2018 | PMID: 29294095 | Materials and methods subsection: Yeast strains and growth |
| Commercial assay or kit | TruSeq ChIP Sample Preparation Kit | Illumina | IP-202-9001DOC | |
| Antibody, rabbit polyclonal | Anti-RNA polymerase II CTD repeat YSPTSPS (phospho S5) | Abcam | ab5131 | Rabbit polyclonal; (1 ug per ChIP (50 ml OD = O0.5) |
| Antibody, rabbit polyclonal | Anti-Rap1 | N/A *Schawalder et al., 2004* (PMID:15616569) | RRID: AB_2801428 | Rabbit polyclonal; (5 ul per ChIP (50 ml OD = O0.5) |
| Antibody | Anti-Rpl3 | Provided by *Warner, 1977*. | PMID: 9121443 | Results, *Figure 5*. Mouse monoclonal; 1:10 000 |
| Antibody | Anti-Rpl6 | Provided by O Gadal | | Rabbit polyclonal; 1:10 000 |
| Antibody | Anti-Rps8 | Provided by G Dieci | | Rabbit polyclonal; 1:10 000 |
| Antibody, rabbit polyclonal | antibody Ifh1 | N/A (*Knight et al., 2014*) PMID: 25085421 | AB_2801429 | Rabbit polyclonal (2 ul per ChIP (50 ml OD = O0.5) |
| Antibody, rabbit polyclonal | Antibody Fhl1 | N/A (*Knight et al., 2014*) PMID: 25085421 | AB_2801431 | Rabbit polyclonal (2 ul per ChIP (50 ml OD = O0.5) |
| Strain, strain background | *Saccharomyces cerevisiae*, W303 | W303: *MATa/MATα leu2-3,112 trp1-1 can1-100 ura3-1 ade2-1 his3-11,15* | Thomas & Rothstein, 1989. PMID: 2645056. Experimental procedures, Strains. Table 1. | See *Supplementary file 7* |
| Other | Primary sequence files | GEO | accession number GSE125226 | Materials and methods subsection: Plasmid construction |
| Recombinant DNA reagent | pFA6a-link-GFPEnvy-SpHis5 (plasmid) | Addgene | RRID: Addgene_60782 | Materials and methods subsection: Plasmid construction |
| Recombinant DNA reagent | RPL25-envy GFP-plasmid | This paper | Plasmid #1037 | Materials and methods subsection: Plasmid construction |
| Recombinant DNA reagent | pRS315-RPL25-eGFP | *Milkereit et al. (2001)* PMID: 11313466 | | Materials and methods subsection: Plasmid constructions |

## ChIP-Seq

Cultures of 50 mL in YPAD were collected at $OD_{600}$0.4–0.8 for each condition. The cells were cross-linked with 1% formaldehyde for 5 min at room temperature and quenched by adding 125 mM glycine for 5 min at room temperature. Cells were washed with ice-cold HBS and resuspended in 3.6 mL of ChIP lysis buffer (50 mM HEPES-Na pH 7.5, 140 mM NaCl, 1 mM EDTA, 1% NP-40, 0.1% sodium deoxycholate) supplemented with 1 mM PMSF and 1x protease inhibitor cocktail (Roche).

Samples were aliquoted in Eppendorf tubes and frozen. After thawing, the cells were broken using Zirconia/Silica beads (BioSpec). The lysate was spun at 13,000 rpm for 30 min at 4°C and the pellet was resuspended in 300 µl ChIP lysis buffer + 1 mM PMSF and sonicated for 15 min (30 s ON - 60 s OFF) in a Bioruptor (Diagenode). The lysate was spun at 7000 rpm for 15 min at 4°C. Antibody (1 µg / 300 µL of lysate, Abcam ab5131) was added to the supernatant and incubated for 1 hr at 4°C. Magnetic beads were washed three times with PBS plus 0.5% BSA and added to the lysates (30 µL of beads/300 µL of lysate). The samples were incubated for 2 hr at 4°C. The beads were washed twice with (50 mM HEPES-Na pH 7.5, 140 mM NaCl, 1 mM EDTA, 0.03% SDS), once with AT2 buffer (50 mM HEPES-Na pH 7.5, 1 M NaCl, 1 mM EDTA), once with AT3 buffer (20 mM Tris-Cl pH 7.5, 250 mM LiCl, 1 mM EDTA, 0.5% NP-40, 0.5% sodium deoxycholate) and twice with TE. The chromatin was eluted from the beads by resuspension in TE + 1% SDS and incubation at 65°C for 10 min. The eluate was transferred to an Eppendorf tube and incubated overnight at 65°C to reverse the crosslinks. The DNA was purified using High Pure PCR Cleanup Micro Kit (Roche). DNA libraries were prepared using TruSeq ChIP Sample Preparation Kit (Illumina) according to manufacturer's instructions. The libraries were sequenced using an Illumina HiSeq 2500 and the reads were mapped to the sacCer3 genome assembly using HTSStation (shift = 150 bp, extension = 50 bp; *David et al., 2014*). To compare depleted versus non-depleted cells, we divided the signal from the + auxin and/or rapamycin and/or cycloheximide samples by the signal from the – auxin and/or rapamycin and/or cycloheximide (vehicle) samples and log2 transformed this value. All data from publicly available databases were mapped using HTS Station (http://htsstation.epfl.ch; *David et al., 2014*).

## Yeast strains, primer DNAs and cell growth

Strains used in this study are listed in *Supplementary file 7*. For ChIP-qPCR, the primer sequences used a listed in *Supplementary file 8*. Experiments were typically performed with log phase cells harvested between $OD_{600}$ 0.4 and 0.8. Anchor-away of FRB-tagged proteins was induced by the addition of rapamycin (1 mg/ml of 90% ethanol/10% Tween stock solution) to a final concentration of 1 µg/ml (*Haruki et al., 2008*). Depletion of AID-tagged protein was induced by the addition of auxin (3-indoleacetic acid) at 500 µM final concentration. Arrest of translation was induced by the addition of cycloheximide to a final concentration of 25 µg/ml. Cells are treated with diazaborine to a final concentration of 50 ug/ml.

## Fluorescence microscopy

Cells were grown overnight at 30°C in SC medium (0.67% nitrogen base without amino acids (BD), 2% dextrose supplemented with amino acids mixture (AA mixture; Bio101), adenine, and uracil). Cells were diluted and were harvested when $OD_{600}$ reached 0.4. Cells were spread on slides coated with an SC medium patch containing 2% glucose. Stacked images were recorded (Intelligent Imaging Innovations) at a spinning disc confocal inverted microscope (Leica DMIRE2) using the 100x oil objective and an Evolve EMCCD Camera (Photometrics).

## Insoluble fraction purification and mass spectrometry

Isolation of protein aggregates from yeast cells was performed as described previously (*Koplin et al., 2010*) with slight modifications. 50 $OD_{600}$ units (50 ml) of exponentially growing cells were harvested, and cell pellets were frozen in liquid N2. The cell pellets were resuspended in 1 ml lysis buffer (20 mM Na-phosphate pH 6.8, 10 mM DTT, 1 mM EDTA, 0.1% Tween, 1 mM PMSF, protease inhibitor cocktail and 100 units/ml zymolyase) and incubated at 30° C for 30 min. Chilled samples were treated by tip sonication (20%, 10 s, 2x) and centrifuged for 20 min at 600 g at 4°C. Aggregated proteins were pelleted at 16,000 g for 20 min at 4°C. After removing supernatants, insoluble proteins were washed once with Wash I buffer (20 mM Na-phosphate pH 6.8, 500 mM NaCl, 5 mM EDTA, 2% NP-40, 1 mM PMSF, and protease inhibitor cocktail), and centrifuged at 16,000 g for 20 min at 4°C. Insoluble proteins were washed with Wash II buffer (20 mM Na-phosphate pH 6.8, ice-cold), pelleted and sonicated (2x for 10 s) in 40 µl of Wash II buffer. For analysis by SDS-PAGE (4–12% acrylamide) and subsequent western blotting, proteins were first boiled in Laemmli buffer. 1x of the total cell lysate and 25x of the insoluble pellet fraction were separated and analyzed by Coomassie Blue staining or immunoblotting. Proteins were identified by shotgun mass spectrometry analysis at the Functional Genomics Center Zurich (ETH, Zurich) following TCA

precipitation (20%) and acetone washing, according to posted procedures. Database searches were performed by using the Mascot (SwissProt, all species; SwissProt, yeast) search program, using very stringent settings in Scaffold (1% protein FDR, a minimum of 2 peptides per protein, 0.1% peptide FDR).

## Polysome gradients

Yeast cells growing exponentially were treated or not with auxin for 20 min. 50 µg/mL cycloheximide (Sigma) was added directly to the culture medium. Cells were collected by centrifugation, rinsed with buffer K [20 mM Tris-HCl pH 7.4, 50 mM KCl, 10 mM $MgCl_2$] supplemented with 50 µg/mL cycloheximide and collected again by centrifugation. Dry pellets were resuspended with approximately one volume of ice-cold buffer K supplemented with 1 mM DTT, 1 × Complete EDTA-free protease inhibitor cocktail (Roche), 0.1 U/µL RNasin (Promega) and 50 µg/mL cycloheximide. About 250 µL of ice-cold glass beads (Sigma) were added to 500 µL aliquots of the resuspended cells and cells were broken by vigorous shaking, three times 2 min, separated by 2 min incubations on ice. Extracts were clarified through two successive centrifugations at 13,000 rpm and 4°C for 5 min and quantified by measuring absorbance at 260 nm. About 30 $A_{260}$ units were loaded onto 10–50% sucrose gradients in buffer K, and then centrifuged for 150 min at 39,000 rpm and 4°C in an Optima L-100XP Ultracentrifuge (Beckman-Coulter) using a SW41Ti rotor without brake. Following centrifugation, 18 fractions of 500 µl each were collected from the top of the gradients with a Foxy Jr. apparatus (Teledyne ISCO). The absorbance at 254 nm was measured during collection with a UA-6 device (Teledyne ISCO).

## Pulse labeling, RNA extraction and Northern hybridization

Metabolic labeling of pre-rRNAs was performed as previously described (*Tollervey et al., 1993*) with the following modifications. Strains were grown in synthetic glucose medium lacking adenine to an $OD_{600}$ of 0.8. Auxin (0.5 mM) was next added to the cultures and cells were labeled for 2 min with [2,8-$^3$H]-adenine (NET06300 Perkin Elmer) at 0, 10, 20 and 30 min following the addition of auxin. Cell pellets were frozen in liquid nitrogen. RNA extractions and Northern hybridizations were performed as previously described (*Beltrame and Tollervey, 1992*). For high-molecular-weight RNA analysis, 2 µg of total RNA were glyoxal denatured and resolved on a 1.2% agarose gel. Note that Northern hybridization was performed on [2,8-$^3$H]-adenine labeled RNA. The membrane was first exposed to reveal neo-synthetized transcripts, and subsequent Northern hybridization revealed rRNA transcript abundance.

## Data availability

Read counts for all RNAPII ChIP-seq experiments (integrated counts over the complete open reading frame of all protein-coding genes) are given in *Supplementary file 1–6*. Primary processed sequence files will be made available at Gene Expression Omnibus (GEO accession number GSE125226).

## Acknowledgements

We thank Mylène Docquier and the Genomics Platform of iGE3 at the University of Geneva (https://ige3.genomics.unige.ch/) for high-throughput sequencing services, the Functional Genomics Center Zurich (ETH, Zurich) for mass spectrometry analysis, Prof. Helmut Bergler (Karl-Franzens-Universität, Graz, Austria) for his generous gift of diazaborine, Lyudmil Raykov and the Bioimaging Center at the Faculty of Sciences, University of Geneva (http://bioimaging.unige.ch/) for help with confocal microscopy, Nicolas Roggli for expert assistance with data presentation and artwork, Philipp Milkereit and Robbie Loewith for enlightening discussions and all members of the Shore lab for comments and discussions throughout the course of this work. BA acknowledges support from a long-term EMBO postdoctoral fellowship in the early phases of this work. MJB was supported in part by an iGE3 Ph. D. student fellowship. DS acknowledges funding from the Swiss National Science Foundation (grant number 31003A_170153) and the Republic and Canton of Geneva.

## Additional information

### Funding

| Funder | Grant reference number | Author |
|---|---|---|
| Schweizerischer Nationalfonds zur Förderung der Wissenschaftlichen Forschung | 31003A_170153 | David Shore |

The funders had no role in study design, data collection and interpretation, or the decision to submit the work for publication.

### Author contributions

Benjamin Albert, Conceptualization, Data curation, Formal analysis, Investigation, Writing—original draft; Isabelle C Kos-Braun, Christophe Dez, Xu Zhang, Investigation, Substantial contributions to conception and design, Acquisition of data, or analysis and interpretation of data, Made final approval of the version to be published; Anthony K Henras, Investigation, Writing—review and editing; Maria Paula Rueda, Investigation, substantial contributions to conception and design, Acquisition of data, or analysis and interpretation of data, Made final approval of the version to be published; Olivier Gadal, Conceptualization, Formal analysis, Supervision, Writing—review and editing; Martin Kos, Data curation, Supervision, Funding acquisition, Investigation, Writing—review and editing; David Shore, Conceptualization, Formal analysis, Supervision, Funding acquisition, Project administration, Writing—review and editing

### Author ORCIDs

Olivier Gadal http://orcid.org/0000-0001-9421-0831
Martin Kos https://orcid.org/0000-0002-3337-9681
David Shore https://orcid.org/0000-0002-9859-143X

### Decision letter and Author response

Decision letter https://doi.org/10.7554/eLife.45002.028
Author response https://doi.org/10.7554/eLife.45002.029

## Additional files

### Supplementary files

• Supplementary file 1. Rpb1 ChIP-seq at the indicated times following either auxin or vehicle addition to the media in Top1-AID, Top2-AID or Top1-AID Top2-AID strain.
DOI: https://doi.org/10.7554/eLife.45002.016

• Supplementary file 2. Rpb1 ChIP-seq at the indicated times following either auxin (Aux), vehicle (veh), diazaborine (Diaz), or cycloheximide (CHX) addition to the media.
DOI: https://doi.org/10.7554/eLife.45002.017

• Supplementary file 3. Number of peptides purified in an insoluble fraction from Top1/2-AID cells treated for 20 min with either auxin or vehicle.
DOI: https://doi.org/10.7554/eLife.45002.018

• Supplementary file 4. Rpb1 ChIP-seq in Top1-AID Ifh1-FRB or Top1-AID Top2-AID Ifh1-FRB cells following either auxin, auxin plus rapamycin, or rapamycin addition to the media.
DOI: https://doi.org/10.7554/eLife.45002.019

• Supplementary file 5. Rpb1 ChIP-seq in Top1-AID or Top1/2-AID cells following either auxin, auxin plus cycloheximide, or vehicle addition to the media.
DOI: https://doi.org/10.7554/eLife.45002.020

• Supplementary file 6. Rpb1 ChIP-seq in wild type cells following either Vehicle (Veh) or Rapamycin (Rap) addition, Cycloheximide pre-treatment (CHX), or Heat Shock at 40˚C (Heat Shock).
DOI: https://doi.org/10.7554/eLife.45002.021

• Supplementary file 7. Yeast strains used in this study.

DOI: https://doi.org/10.7554/eLife.45002.022

• Supplementary file 8. DNA primers used in this study.
DOI: https://doi.org/10.7554/eLife.45002.023
• Transparent reporting form
DOI: https://doi.org/10.7554/eLife.45002.024

## Data availability

Sequencing data have been deposited in GEO under accession code GSE125226. Previously published data were used from Supplementary file 3 of Sung et al. 2016, eLife (https://elifesciences.org/articles/19105/figures#SD7data) and Supplemental Table S3 from Sung et al. 2016, Mol Biol Cell (supp_E16-05-0290v1_mc-E16-05-0290-s06.xlsx).

The following dataset was generated:

| Author(s) | Year | Dataset title | Dataset URL | Database and Identifier |
|---|---|---|---|---|
| Albert B, Kos-Braun IC, Henras AK, Dez C, Paula Rueda M, Zhang X, Gadal O, Kos M, Shore D | 2019 | Sequencing data from A ribosome assembly stress response regulates transcription to maintain proteome homeostasis | https://www.ncbi.nlm.nih.gov/geo/query/acc.cgi?acc=GSE125226 | NCBI Gene Expression Omnibus, GSE125226 |

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
