## [Decision Letter]

Thank you for submitting your article "A ribosome assembly stress response regulates transcription to maintain proteome homeostasis" for consideration by *eLife*. Your article has been reviewed by three peer reviewers, one of whom is a member of our Board of Reviewing Editors, and the evaluation has been overseen by Naama Barkai as the Senior Editor. The following individuals involved in review of your submission have agreed to reveal their identity: John Woolford (Reviewer #2); Marlene Oeffinger (Reviewer #3).

The reviewers have discussed the reviews with one another and the Reviewing Editor has drafted this decision to help you prepare a revised submission.

Summary:

This report shows that depletion of Top1, which leads to arrest of Pol I elongation, evokes a specific induction of the Hsf1 regulon, and not the general ESR, which can be blocked by anchor away of Hsf1; and also evokes the rapid downregulation of nearly all ribosome protein genes (RPGs), which can be attributed to loss of the transcription factor Ifh1 from RPG promoters. They term this response RASTR for Ribosome Assembly Stress Response. Evidence is presented that the signal for both responses, Hsf1 target induction and RPG repression, is a defect in 60S ribosome biogenesis associated with accumulation of nuclear aggregates of unassembled RPs, which sequester Ifh1 from RPG promoters. They show that blocking new synthesis of RPs by anchor away of Ifh1 blocks induction of Hsf1 targets on depletion of Top1. Moreover, they show that cycloheximide (CHX) treatment blocks both aspects of RASTR, presumably by preventing accumulation of newly synthesized RPs in nuclear aggregates capable of sequestering Ifh1, (although this effect of CHX on aggregation of RPs and Ifh1 was not directly demonstrated). CHX also appears to block the RASTR that is triggered immediately by Rapamycin presumably owing to an immediate inhibition of rRNA processing before the onset of RP repression, leading to accumulation of RP aggregates that sequester Ifh1. However, again, this interpretation was not supported by direct evidence of nuclear aggregates of RPs and Ifh1 at early, but not later, times of Rapamycin treatment, which are eliminated by CHX treatment. The authors conclude that unassembled RPs in nuclear aggregates serve to activate the Hsf1 regulon and down-regulate RPG transcription by sequestering Ifh1 as a means of tightly regulating RP expression and responding to the toxicity of RP aggregates, which functions both in normal cells and during conditions of unbalanced RP and rRNA production when ribosome biogenesis is impaired, when TORC1 is acutely inhibited by Rapamycin, and during heat-shock.

Essential revisions:

Additional experiments are required to establish that the accumulation of ribosomal proteins and Ifh1 in nuclear aggregates is diminished in cells treated with cycloheximide (CHX) to justify the interpretations of multiple experiments in Figures 5-6, in which CHX is found to block RASTR in response to Top1 depletion or at the early time points of rapamycin treatment. The authors should also confirm that Ifh1 promoter occupancy is reduced in tom1 cells to bolster their interpretation of the genetic data in Figure 3D. Revisions of text are also needed throughout the paper to better explain the rationale behind certain experiments or their design, and to expand the Discussion to consider the signals for triggering RASTR (does it involve unassembled 40S proteins, is it limited to the early stages of ribosome assembly), the conditions that lead to cessation of the response, and situations in WT cells in which it would be triggered (all described in the major comments of reviewer #2 and #3).

Reviewer #1:

This report shows that depletion of Top1, which leads to arrest of Pol I elongation evokes a specific induction of the Hsf1 regulon, and not the general ESR, which is blocked by anchor away of Hsf1; and also to rapid downregulation of nearly all ribosome protein genes (RPGs), which can be attributed to loss of the transcription factor Ifh1 from RPG promoters. They term this response RASTR for Ribosome Assembly Stress Response. Evidence is presented that the signal for both responses, Hsf1 target induction and RPG repression, is a defect in 60S ribosome biogenesis associated with accumulation of nuclear aggregates of unassembled RPs, which sequester Ifh1 from RPG promoters. Unlike in cells treated with Rapamycin, in which Ifh1 also is sequestered, the removal of Ifh1 from RPG promoters in RASTR does not required the Ifh1 C-terminal domain. They show that blocking new synthesis of RPs by anchor away of Ifh1 blocks induction of Hsf1 targets on depletion of Top1. Moreover, they show that cycloheximide (CHX) treatment blocks both aspects of RASTR, presumably by preventing accumulation of newly synthesized RPs in nuclear aggregates capable of sequestering Ifh1, although this effect of CHX on aggregation of RPs and Ifh1 is not directly demonstrated. CHX also appears to block the RASTR that is triggered immediately by Rapamycin presumably owing to an immediate inhibition of rRNA processing before the onset of RP repression, leading to accumulation of RP aggregates that sequester Ifh1. However, again, this interpretation was not supported by direct evidence of nuclear aggregates of RPs and Ifh1 at early, but not later, times of Rapamycin treatment, which are eliminated by CHX treatment. The authors conclude that unassembled RPs in nuclear aggregates serve to activate the Hsf1 regulon and down-regulate RPG transcription by sequestering Ifh1 as a means of tightly regulating RP expression and responding to the toxicity of RP aggregates, which functions both in normal cells and during conditions of unbalanced RP and rRNA production when ribosome biogenesis is impaired, TORC1 is acutely inhibited by Rapamycin, and in heat-shock.

The findings in this paper are significant in showing specific induction of the Hsf1 regulon, and not the general ESR, when ribosome biogenesis is impaired either by depleting Top1, known already and confirmed here to arrest Pol I elongation, or by interfering with 60S biogenesis in different ways; and in describing a mechanism for the concurrent reduction in RP transcription via sequestration of Ifh1 in nuclear aggregates of unassembled ribosomal proteins.

- It was felt that direct evidence is needed that the accumulation of RPs and Ifh1 in nuclear aggregates is diminished in cells treated with cycloheximide (CHX) to justify their interpretations of the multiple experiments in Figures 5-6 in which CHX is found to block the RASTR in response to Top1 depletion or at the early time points of rapamycin treatment.

- The authors should also show that Ifh1 promoter occupancy is reduced in tom1 cells to bolster their interpretation of the genetic data in Figure 3D.

Reviewer #2:

This is an interesting and timely manuscript describing discovery of the so-called Ribosome Assembly Stress Response (RASTR) pathway: When ribosome assembly is blocked, transcription by Hsf1 to activate protein homeostasis genes (encoding chaperones, aggregates, and proteasomal proteins) is upregulated, and ribosomal protein gene transcription is down-regulated. Further experiments explain how this occurs. When ribosome assembly is blocked by aborting rRNA transcription or preventing assembly factor function, excess unassembled ribosomal proteins aggregate. This leads to sequestration of Hsp70, an inhibitor of Hsf1, as well as Ifh, which is necessary for transcription of RP genes.

The RASTR mediated pathway is rapid and apparently short-lived, activated within 5 minor aborting assembly, but returning to previous levels of expression of r protein proteostasis genes within an hour. It is not yet known whether r proteins, Ifh1, and Hsp70 are present in separate pools of aggregates, or whether or how they traffic into or out of aggregates.

Because ribosomal proteins are synthesized at such high levels in rapidly growing cells, RASTR represents a significant mechanism to prevent proteostasis when assembly is imbalanced by any means.

Some data are presented to distinguish RASTR from other stress response pathways, and to distinguish long term responses from the short term RASTR pathway. I found this portion of the Results and Discussion difficult to follow at times. The manuscript could perhaps be made more clear, especially for a general readership, by including a short paragraph early in the Discussion clearly outlining in one place the different pathways. To what exactly might each pathway respond and why? For example, what are general means by which imbalance in synthesis of ribosome components could be created in wild type cells (or by mutation?) and what then enables a return to normal synthesis once RASTR is inactivated? Is it diminished pools of aggregates? Then what about the primary triggers for the pathway? It also might be worth repeating here exactly where and how the CURI complex fits in to these effects.

Reviewer #3:

In the submitted manuscript by Albert et al., the authors identify the Ribosomal Assembly Stress Response (RASTR), a novel regulatory pathway that responds to excess numbers of RPs and their aggregation, and allows a simultaneous up-regulation of protein homeostasis genes and down-regulation of RP genes upon disruption of ribosome biogenesis at different stages, providing a mechanism for the cell to counteract RP proteotoxicity.

Overall, the data presented here is very interesting and experimentally sound. The data is mostly well-presented and the manuscript generally concise; however, a few points should be addressed by the authors.

- Throughout, the authors focus on the accumulation and fate of large subunit RPs and in Figure 2C and D, only large subunit RPs are shown by western blot. Have the authors looked at small subunit RPs? Given that the stresses shown here affect pre-rRNA transcription and assembly of RPs with pre-rRNA is co-transcriptional, the sRPs should also be affected and involved in the RASTR. The authors should address this point in their Discussion at the very least.

- The authors selected Utp8 and Utp13 to mimic a ribosome assembly stress response as observed with Top1 and Top1/2 deletion; why were these two proteins chosen specifically? This should be explained for an audience less familiar with ribosome biogenesis.

- Along these lines, the authors say that disruption of ribosome biogenesis at several levels – pre-rRNA transcription/processing, assembly – will activate RASTR. What about later stages in assembly/processing than the one tested in the 90S complex? Is the response related to early RiBi pathways steps?

- Figure 2—figure supplement 1D: tae2-Δ cells – the figure/experiment described in the text does not match the panel/legend – please make sure it is the correct data panel or labeled to be made clearer to the reader.

- While the manuscript and the logic of experiments is in most cases obvious to a reader within the immediate field, it may be less so to a general/non-ribosome biogenesis readership. I would suggest the authors be more liberal and expansive with some of their explanations, one example being the choice of Utp8 and Utp13, but there are others throughout.

- Figure 2A: labels for the northern probe(s) used should be added as well as labels for the precursors visualized.

[Editors' note: further revisions were requested prior to acceptance, as described below.]

Thank you for resubmitting your work entitled "A ribosome assembly stress response regulates transcription to maintain proteome homeostasis" for further consideration at *eLife*. Your revised article has been favorably evaluated by Naama Barkai as the Senior Editor and a Reviewing Editor.

The manuscript has been improved substantially but there are a few remaining issues that need to be addressed before acceptance, as outlined below:

1) The new experiment presented in Figure 3D lacks the WT TOM1 control strain needed to establish that the Ifh1 foci occur only in the tom1 mutant cells.

2) It appears that the data are mislabeled in the new Figure 5—figure supplement 1E-F.

3) Please re-proof all new figures and revised text for correctness.

---

## [Author Response]

Reviewer #1:[…] – It was felt that direct evidence is needed that the accumulation of RPs and Ifh1 in nuclear aggregates is diminished in cells treated with cycloheximide (CHX) to justify their interpretations of the multiple experiments in Figures 5-6 in which CHX is found to block the RASTR in response to Top1 depletion or at the early time points of rapamycin treatment.

We agree with the reviewer that these data are important for interpretation of the experiments shown in Figures 5 and 6. To address this issue, we first point out more clearly in the revised text that the Deshaies lab (Sung et al., 2016) have already shown by MS analysis that RP aggregation following proteasome inhibition is strongly diminished upon cycloheximide (CHX) addition. Since the aggregation assay that they used didn’t allow them to discriminate between cytosolic and nuclear aggregates we decided to use an alternative approach (live cell imaging) to observe accumulation of RPs and Ifh1 in protein aggregates. To this end, we expressed Super-Fold GFP-tagged versions of Ifh1 and Rpl25 and observed their aggregation (foci formation) in the presence or absence of CHX. Significantly, we found that Ifh1 forms small punctate structures in a *tom1-Δ* strain (new Figure 3D). Remarkably, these nuclear structures rapidly diminish (20 min) after CHX treatment (new Figure 5—figure supplement 1B). Furthermore, we have now added data showing that the accumulation of Ifh1 in nuclear aggregates observed during RASTR (old Figure 3C) is abolished in the presence of cycloheximide (CHX) (new Figure 5G),in agreement with the proposed model, now in Figure 8. Lastly, we observed during RASTR activation that Rpl25-eGFP rapidly forms a nuclear structure that is not observed in the presence of CHX (new Figure 5F). Taken together, we believe that our new results provide very clear evidence that CHX blocks (or strongly diminishes) the formation of both RP and Ifh1 aggregates following RASTR activation.

- The authors should also show that Ifh1 promoter occupancy is reduced in tom1 cells to bolster their interpretation of the genetic data in Figure 3D.

As suggested by the reviewer, we performed Ifh1 ChIP in *tom1-*Δ cells, which showed that Ifh1 binding at *RPL30* appears to be reduced compared to that in *TOM1* cells (new Figure 5—figure supplement 1C). However, this reduction is small, and we are hesitant to place too much weight on quantitative comparisons of ChIP enrichment in different strains. Nevertheless, this small difference in Ifh1 binding is in agreement with the near wild-type growth rate of the *tom1-Δ* strain, suggesting that the level of Ifh1 promoter occupancy in *tom1-Δ* remains sufficient to promote near normal RP gene transcription. Importantly, though, cycloheximide treatment triggers a significant increase in Ifh1 promoter binding in *tom1-Δ* strains compared to wild type (new Figure 5—figure supplement 1C), consistent with the disappearance of Ifh1 nuclear aggregates upon CHX treatment in the *tom1-Δ* strain (new Figure 5—figure supplement 1B). These data support the idea that *tom1-Δ* does indeed increase the amount of Ifh1 trapped in an insoluble nuclear fraction that can be rapidly released upon CHX treatment to increase binding at RP promoters.

Lastly, to exclude the possibility that the genetic interaction between *TOM1* and the hypo-morphic *ifh1-AA* allele (shown previously in Figure 3D) could be linked to the growth defect of this mutation, we examined another mutated allele of *IFH1 (ifh1-6*) that triggers a similar growth defect (new Figure 3E).

Importantly, the ifh1-6 mutant protein remains bound at high levels to RP gene promoters even under stress conditions, whereas ifh1-AA interacts weakly with RP gene promoters, even under optimal growth conditions (Albert et al., 2016). Remarkably, *tom1-Δ* is synthetically lethal only with ifh1-AA (new Figure 3E), supporting the notion that this genetic interaction is directly linked to the ability of Ifh1 to bind RP gene promoters. Altogether, we believe that these data strongly support the idea that one part of the pool of Ifh1 is trapped in an insoluble nuclear fraction in absence of Tom1 and prevented from binding to RP gene promoters.

Reviewer #2:This is an interesting and timely manuscript describing discovery of the so-called Ribosome Assembly Stress Response (RASTR) pathway: When ribosome assembly is blocked, transcription by Hsf1 to activate protein homeostasis genes (encoding chaperones, aggregates, and proteasomal proteins) is upregulated, and ribosomal protein gene transcription is down-regulated. Further experiments explain how this occurs. When ribosome assembly is blocked by aborting rRNA transcription or preventing assembly factor function, excess unassembled ribosomal proteins aggregate. This leads to sequestration of Hsp70, an inhibitor of Hsf1, as well as Ifh, which is necessary for transcription of RP genes.The RASTR mediated pathway is rapid and apparently short-lived, activated within 5 minor aborting assembly, but returning to previous levels of expression of r protein proteostasis genes within an hour. It is not yet known whether r proteins, Ifh1, and Hsp70 are present in separate pools of aggregates, or whether or how they traffic into or out of aggregates.Because ribosomal proteins are synthesized at such high levels in rapidly growing cells, RASTR represents a significant mechanism to prevent proteostasis when assembly is imbalanced by any means.Some data are presented to distinguish RASTR from other stress response pathways, and to distinguish long term responses from the short term RASTR pathway. I found this portion of the Results and Discussion difficult to follow at times. The manuscript could perhaps be made more clear, especially for a general readership, by including a short paragraph early in the Discussion clearly outlining in one place the different pathways. To what exactly might each pathway respond and why? For example, what are general means by which imbalance in synthesis of ribosome components could be created in wild type cells (or by mutation?).

These are excellent questions, and we agree that the original manuscript was overly concise, to the point where even well-informed general readers might have difficulty following the arguments. To improve clarity, we have expanded both the Discussion and Introduction, emphasizing more the identity and role of the CURI complex in balancing rRNA and RP production at extended periods following stress (TORC1 inhibition) and pointing out that RASTR is a mechanistically distinct and faster acting response. We also include speculation regarding the physiological and genetic perturbations that might activate these two different responses.

What then enables a return to normal synthesis once RASTR is inactivated? Is it diminished pools of aggregates?

As pointed out above, our new data strongly suggest that the return to normal RP and Hsf1 target gene expression is promoted by the diminution of pools of aggregates, of both Ifh1 and RPs. This is now clearly stated in the text.

Then what about the primary triggers for the pathway? It also might be worth repeating here exactly where and how the CURI complex fits in to these effects.

This is a good point. We add speculation in the Discussion to the effect that the long-term regulation through the CURI complex may be driven by the release of its two rRNA processing components, Utp22 and Rrp7 from nascent rDNA, freeing them for interaction with Ifh1.

Reviewer #3:In the submitted manuscript by Albert et al., the authors identify the Ribosomal Assembly Stress Response (RASTR), a novel regulatory pathway that responds to excess numbers of RPs and their aggregation, and allows a simultaneous up-regulation of protein homeostasis genes and down-regulation of RP genes upon disruption of ribosome biogenesis at different stages, providing a mechanism for the cell to counteract RP proteotoxicity.Overall, the data presented here is very interesting and experimentally sound. The data is mostly well-presented and the manuscript generally concise; however, a few points should be addressed by the authors.- Throughout, the authors focus on the accumulation and fate of large subunit RPs and in Figure 2C and D, only large subunit RPs are shown by western blot. Have the authors looked at small subunit RPs? Given that the stresses shown here affect pre-rRNA transcription and assembly of RPs with pre-rRNA is co-transcriptional, the sRPs should also be affected and involved in the RASTR. The authors should address this point in their Discussion at the very least.

This is a good point, and we agree that sRPs should be affected by and involved in RASTR. Consistent with this, we detect by mass spectrometry both large and small subunit RPs (43 Rpl and 30 Rps proteins) in the insoluble fraction following topoisomerases depletion (new Figure 3—figure supplement 1B) and we detect accumulation of a small subunit protein, Rps8, in trailing fractions of polysome gradients (new Figure 2C). We thank the reviewer for pointing this out.

- The authors selected Utp8 and Utp13 to mimic a ribosome assembly stress response as observed with Top1 and Top1/2 deletion; why were these two proteins chosen specifically? This should be explained for an audience less familiar with ribosome biogenesis.

We agree with the reviewer. We now explain in the revised text why these factors were selected. Please see our reply to comment #4 of the reviewer #2.

- Along these lines, the authors say that disruption of ribosome biogenesis at several levels – pre-rRNA transcription/processing, assembly – will activate RASTR. What about later stages in assembly/processing than the one tested in the 90S complex? Is the response related to early RiBi pathways steps?

We agree with the reviewer that this is an important point. In the manuscript we show that diazaborine, an inhibitor acting at the mid-late steps of 60S biogenesis, activates RASTR, suggesting that this pathway is not limited to early steps. Nevertheless, it’s important to note that blocking maturation of late cytoplasmic large subunit precursors results in a rapid depletion of early acting factors in the nucleolus due to trapping of these factors in the accumulating late populations (Zisser et al. [2018] *NAR* 46: 3140). The failure to recycle these proteins results in a rapid feedback at an early maturation step. We speculate that both early and late stages of assembly will rapidly activate RASTR but acknowledge that further work will be required to nail this down.

- Figure 2—figure supplement 1D: tae2-Δ cells – the figure/experiment described in the text does not match the panel/legend – please make sure it is the correct data panel or labeled to be made clearer to the reader.

We thank the reviewer for catching this mistake. The text and figure now describe Rpb1 ChIP of two Hsf1 target genes and two RP genes following Top1/2 degradation in a *tae2-*Δ background. The data (qPCR ChIP) show that both sets of genes respond as in *TAE2* wild-type cells, ruling out a role for RQC in RASTR (see new Figure 2—figure supplement 1D).

- While the manuscript and the logic of experiments is in most cases obvious to a reader within the immediate field, it may be less so to a general/non-ribosome biogenesis readership. I would suggest the authors be more liberal and expansive with some of their explanations, one example being the choice of Utp8 and Utp13, but there are others throughout.

We agree with the reviewer that the text is too concise and lacking in detail. We have extensively expanded the text to make both the experiments and their interpretation clearer.

- Figure 2A: labels for the northern probe(s) used should be added as well as labels for the precursors visualized.

We apologize for this omission. Please see reply to the comment #1 of the reviewer #2.

[Editors' note: further revisions were requested prior to acceptance, as described below.]

Thank you for resubmitting your work entitled "A ribosome assembly stress response regulates transcription to maintain proteome homeostasis" for further consideration at eLife. Your revised article has been favorably evaluated by Naama Barkai as the Senior Editor and a Reviewing Editor.The manuscript has been improved substantially but there are a few remaining issues that need to be addressed before acceptance, as outlined below:1) The new experiment presented in Figure 3D lacks the WT TOM1 control strain needed to establish that the Ifh1 foci occur only in the tom1 mutant cells.

We have now included the *TOM1* control in Figure 3D which establishes that Ifh1 does not form foci in wild-type cells. This was apparent from other images, but you’re right that it’s best to show the control directly in Figure 3D. Thanks for pointing this out.

2) It appears that the data are mislabeled in the new Figure 5—figure supplement 1E-F.

You’re absolutely right about the swapped labeling on Figure 5—figure supplement 1E-F, and we thank you for pointing out this mistake, which has now been corrected.

3) Please re-proof all new figures and revised text for correctness.

Checking everything again we realized that there were graphical mistakes in the cartoons of Figures 1, 4 and 5. These have been corrected and we have uploaded the revised figures. The rest of the manuscript and figures have been checked and appear to us to be correct.